# Enhanced neural speech tracking through noise indicates stochastic resonance in humans

Björn Herrmann[1,2]*

[1]Rotman Research Institute, Baycrest Academy for Research and Education, Toronto, Canada; [2]Department of Psychology, University of Toronto, Toronto, Canada

## eLife Assessment

This study presents an **important** contribution to the understanding of neural speech tracking, demonstrating how minimal background noise can enhance the neural tracking of the amplitude-onset envelope. The evidence, through a well-designed series of EEG experiments, is **convincing**. This work will be of interest to auditory scientists, particularly those investigating biological markers of speech processing.

*For correspondence: bherrmann@research.baycrest.org

Competing interest: The author declares that no competing interests exist.

**Abstract** Neural activity in auditory cortex tracks the amplitude-onset envelope of continuous speech, but recent work counterintuitively suggests that neural tracking increases when speech is masked by background noise, despite reduced speech intelligibility. Noise-related amplification could indicate that stochastic resonance – the response facilitation through noise – supports neural speech tracking, but a comprehensive account is lacking. In five human electroencephalography experiments, the current study demonstrates a generalized enhancement of neural speech tracking due to minimal background noise. Results show that (1) neural speech tracking is enhanced for speech masked by background noise at very high signal-to-noise ratios (~30 dB SNR) where speech is highly intelligible; (2) this enhancement is independent of attention; (3) it generalizes across different stationary background maskers, but is strongest for 12-talker babble; and (4) it is present for headphone and free-field listening, suggesting that the neural-tracking enhancement generalizes to real-life listening. The work paints a clear picture that minimal background noise enhances the neural representation of the speech onset-envelope, suggesting that stochastic resonance contributes to neural speech tracking. The work further highlights non-linearities of neural tracking induced by background noise that make its use as a biological marker for speech processing challenging.

## Introduction

Speech in everyday life is often masked by background sound, such as music or speech by other people, making speech comprehension challenging, especially for older adults (*Pichora-Fuller et al., 2016*; *Herrmann and Johnsrude, 2020*). Speech comprehension challenges are a serious barrier to social participation (*Nachtegaal et al., 2009*; *Heffernan et al., 2016*) and can have long-term negative health consequences, such as cognitive decline (*Lin and Albert, 2014*; *Panza et al., 2019*). Understanding how individuals encode speech in the presence of background sound is thus an important area of research and clinical application. One successful approach to characterize speech encoding in the brain is to quantify how well neural activity recorded with electroencephalography (EEG) tracks relevant speech features of continuous speech, such as the amplitude envelope (*Crosse et al., 2016*; *Crosse et al., 2021*). Greater speech tracking has been associated with higher speech intelligibility

(*Ding et al., 2014*; *Vanthornhout et al., 2018*; *Lesenfants et al., 2019*), leading to the suggestion that speech tracking could be a useful clinical biomarker, for example, in individuals with hearing loss (*Gillis et al., 2022*; *Palana et al., 2022*; *Schmitt et al., 2022*). Counterintuitively, however, neural speech tracking has been shown to increase in the presence of background masking sound even at masking levels for which speech intelligibility is decreased (*Yasmin et al., 2023*; *Panela et al., 2024*). The causes of this increase in speech tracking under background masking are unclear.

In many everyday situations, a listener must block out ambient, stationary background noise, such as multi-talker babble in a busy restaurant. For low signal-to-noise ratios (SNRs) between speech and background noise, neural speech tracking decreases relative to high SNRs (*Ding and Simon, 2013*; *Vanthornhout et al., 2018*; *Zou et al., 2019*; *Yasmin et al., 2023*), possibly reflecting the decreased speech understanding under speech masking. In contrast, for speech in moderate background noise, when a listener can understand most words, neural speech tracking can be increased relative to clear speech (*Yasmin et al., 2023*; *Panela et al., 2024*). This has been interpreted to reflect the increased attention required to understand speech (*Hauswald et al., 2022*; *Yasmin et al., 2023*; *Panela et al., 2024*). However, the low-to-moderate SNRs (e.g., <10 dB) typically used to study neural speech tracking require listeners to invest cognitively and thus do not allow distinguishing a cognitive mechanism from the possibility that the noise per se leads to an increase in neural tracking, for example, through stochastic resonance, where background noise amplifies the response of a system to an input (*Kitajo et al., 2007*; *McDonnell and Ward, 2011*; *Krauss et al., 2016*). Examining neural speech tracking under high SNRs (e.g., >20 dB SNR), for which individuals can understand speech with ease, is needed to understand the noise-related tracking enhancement.

A few previous studies suggest that noise per se may be a relevant factor. For example, the neural-tracking increase has been observed for speech masked by multi-talker background babble (*Yasmin et al., 2023*; *Panela et al., 2024*), but appears to be less present for noise that spectrally matches the speech signal (*Ding and Simon, 2013*; *Synigal et al., 2023*), suggesting that the type of noise is important. Other research indicates that neural responses to tone bursts can increase in the presence of minimal noise (i.e., high SNRs) relative to clear conditions (*Alain et al., 2009*; *Alain et al., 2012*; *Alain et al., 2014*), pointing to the critical role of noise in amplifying neural responses independent of speech.

Understanding the relationship between neural speech tracking and background noise is critical because neural tracking is frequently used to investigate consequences of hearing loss for speech processing (*Presacco et al., 2019*; *Decruy et al., 2020*; *Van Hirtum et al., 2023*). Moreover, older adults often exhibit enhanced neural speech tracking (*Presacco et al., 2016*; *Brodbeck et al., 2018b*; *Broderick et al., 2021*; *Panela et al., 2024*) which is thought to be due to a loss of inhibition and increased internal noise (*Zeng, 2013*; *Auerbach et al., 2014*; *Krauss et al., 2016*; *Zeng, 2020*; *Herrmann and Butler, 2021*). The age-related neural-tracking enhancement may be harder to understand if external, sound-based noise also drives increases in neural speech tracking.

The current study comprises five EEG experiments in younger adults that aim to investigate how neural speech tracking is affected by different degrees of background masking (Experiment 1), whether neural-tracking enhancements are due to attention investment (Experiment 2), the generalizability of changes in neural speech tracking for different masker types (Experiments 3 and 4), and whether effects generalize from headphone to free-field listening (Experiment 5). The results point to a highly generalizable enhancement in neural speech tracking at minimal background masking levels that is independent of attention, suggesting that stochastic resonance plays a critical role.

## Results
### Experiment 1: Enhanced neural speech tracking due to minimal background babble

Experiment 1 aimed to investigate the neural tracking of speech in the presence of different degrees of background masking. Participants (N = 22, median age: 23.5 years) listened to ~2 min stories either in quiet (clear) or in the presence of 12-talker background babble at SNRs ranging from highly intelligible (+30 dB SNR) to moderately difficult intelligibility (−2 dB SNR) while EEG was recorded. Previous work demonstrated that speech at SNRs above ~12 dB is as intelligibility as clear speech (*Holder et al., 2018*; *Rowland et al., 2018*; *Spyridakou et al., 2020*; *Irsik et al., 2022*). Participants

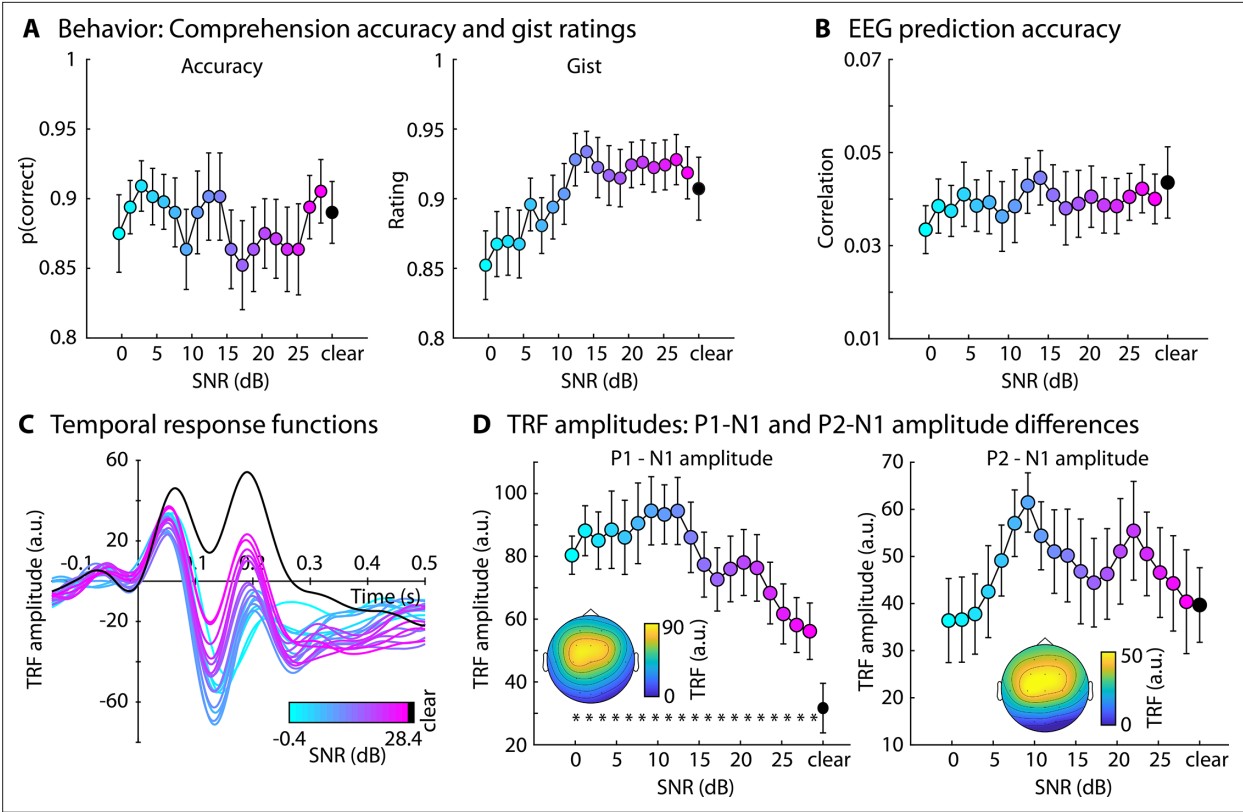

**Figure 1.** Results for Experiment 1 (N=22). (**A**) Accuracy of story comprehension (left) and gist ratings (right). (**B**) Electroencephalography (EEG) prediction accuracy. (**C**) Temporal response functions (TRFs). (**D**) P1-N1 and P2-N1 amplitude difference for different speech-clarity conditions. Topographical distributions reflect the average across all speech-clarity conditions. The black asterisk close to the x-axis indicates a significant difference from a paired t-test relative to the clear condition ($p_{FDR} < 0.05$; false discovery rate [FDR]-thresholded). The absence of an asterisk indicates that there was no significant difference. Error bars reflect the standard error of the mean.

The online version of this article includes the following figure supplement(s) for figure 1:

**Figure supplement 1.** P1-N1 amplitude from temporal response function (TRF) analyses using the amplitude envelope of speech.

**Figure supplement 2.** P1-N1 amplitude from cross-correlations analyses.

answered story comprehension questions and rated gist understanding. Gist ratings for sentences have been shown to correlate highly with intelligibility (word report; *Davis and Johnsrude, 2003*; *Ritz et al., 2022*). The neural tracking of the amplitude-onset envelope of the speech was analyzed using temporal response function (TRF) approaches (*Hertrich et al., 2012*; *Crosse et al., 2016*; *Crosse et al., 2021*).

Comprehension accuracy and gist ratings did not differ between clear speech and any of the SNR levels ($p_{FDR} > 0.05$; thresholded using false discover rate [FDR]; *Benjamini and Hochberg, 1995*; *Genovese et al., 2002*; *Figure 1A*). Because gist ratings appeared to change somewhat with SNR (*Figure 1A*, right), an explorative piece-wise regression was calculated (*McZgee and Carleton, 1970*; *Vieth, 1989*; *Toms and Lesperance, 2003*). The piece-wise regression revealed a breaking point at +15.6 dB SNR, such that gist ratings decreased for +15.6 dB SNR and lower ($t_{21} = 3.008$, p = 0.007, d = 0.641; right to left in *Figure 1A*), whereas gist ratings did not change for +15.6 dB SNR and above ($t_{21} = 0.214$, p = 0.832, d = 0.046).

EEG prediction accuracy did not differ between clear speech and any SNR level ($p_{FDR} > 0.05$; *Figure 1B*). In contrast, the P1-N1 amplitude of the TRF was significantly greater for all SNRs relative to clear speech ($p_{FDR} \leq 0.05$; *Figure 1C, D*; results were comparable when using the amplitude envelope instead of the amplitude-onset envelope, *Figure 1—figure supplement 1*). Explorative piece-wise regression revealed a breaking point at +9.2 dB SNR, showing a significant linear increase in P1-N1 amplitude from +28.4 to +9.2 dB SNR (right to left in *Figure 1D*; $t_{21} = -5.131$, p = $4.4 \cdot 10^{-5}$, d = 1.094), whereas no significant trend was observed for SNRs from about +9.2 to −0.4 dB SNR

($t_{21}$ = 1.001, p = 0.328, $d$ = 0.214). No differences were found for the P2-N1 amplitude ($p_{FDR}$ > 0.05; *Figure 1C, D*).

Experiment 1 replicates the previously observed enhancement in the neural tracking of the speech onset-envelope for speech presented in moderate background babble (*Yasmin et al., 2023*; *Panela et al., 2024*). Critically, Experiment 1 expands the previous work by showing that background babble also leads to enhanced neural speech tracking for very minimal background masking levels (~30 dB SNR). A noise-related enhancement at moderate babble levels (9 dB SNR) has previously been interpreted to result from increased attention or effort to understand speech (*Yasmin et al., 2023*; *Panela et al., 2024*). However, speech for very high SNRs (>15 dB) used in Experiment 1 is highly intelligible and thus should require little or no effort to understand. It is therefore unlikely that increased attention or effort drive the increase in neural speech tracking in background babble. Nonetheless, participants attended to the speech in Experiment 1, and it can thus not be fully excluded that attention investment played a role in the noise-related enhancement.

## Experiment 2: Noise-related enhancement in speech tracking is unrelated to attention

To investigate the role of attention in the noise-related enhancement of neural speech tracking, participants in Experiment 2 (*N* = 22; median age: 23 years) were presented with stories under the same speech-clarity conditions as for Experiment 1 while performing a visual 1-back digit memory task and

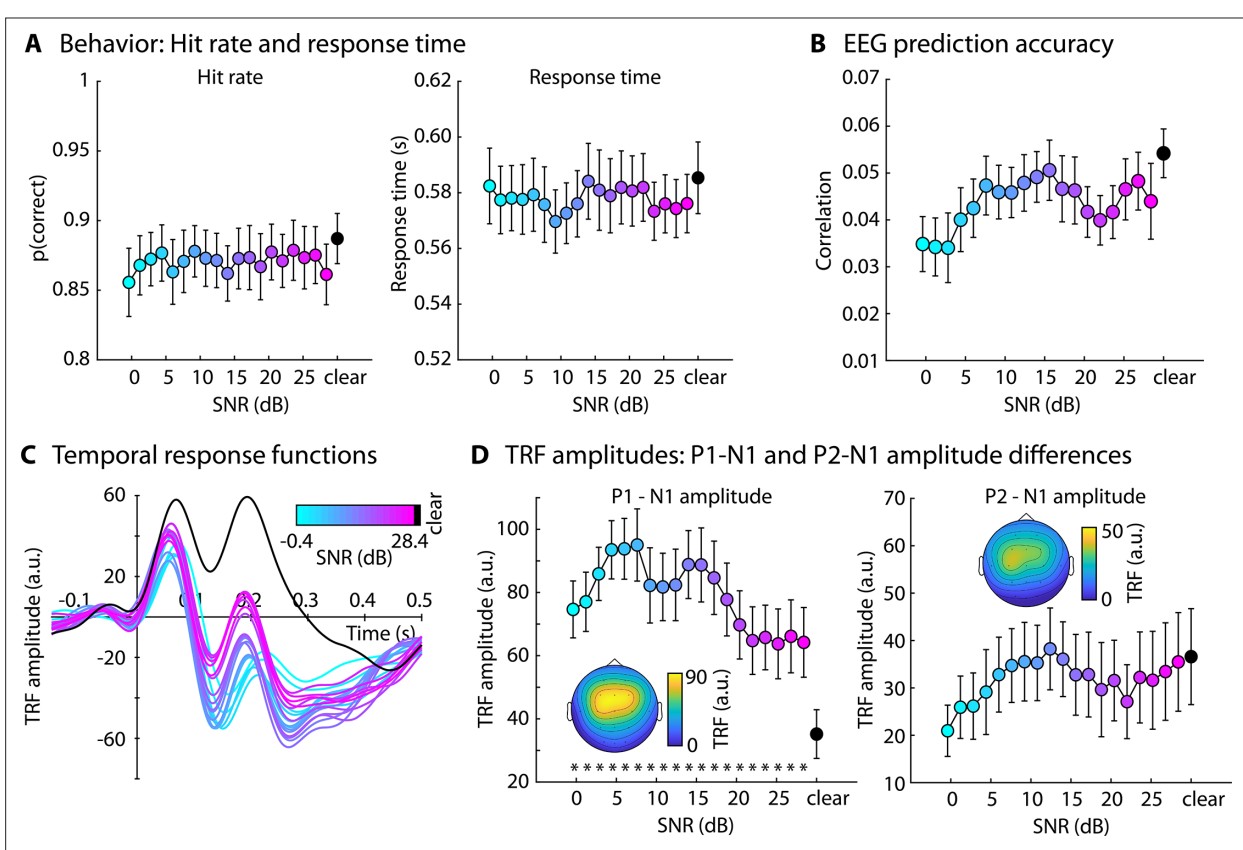

**Figure 2.** Results for Experiment 2 (N=22). (**A**) Hit rate (left) and response times (right) for the visual 1-back task. (**B**) Electroencephalography (EEG) prediction accuracy. (**C**) Temporal response functions (TRFs). (**D**) P1-N1 and P2-N1 amplitude difference for different speech-clarity conditions. Topographical distributions reflect the average across all speech-clarity conditions. The black asterisk close to the x-axis indicates a significant difference from a paired t-test relative to the clear condition ($p_{FDR}$ < 0.05; false discovery rate [FDR]-thresholded). The absence of an asterisk indicates that there was no significant difference. Error bars reflect the standard error of the mean.

The online version of this article includes the following figure supplement(s) for figure 2:

**Figure supplement 1.** Relationship between visual performance and the noise-related enhancement of the P1-N1 amplitude in Experiment 2.

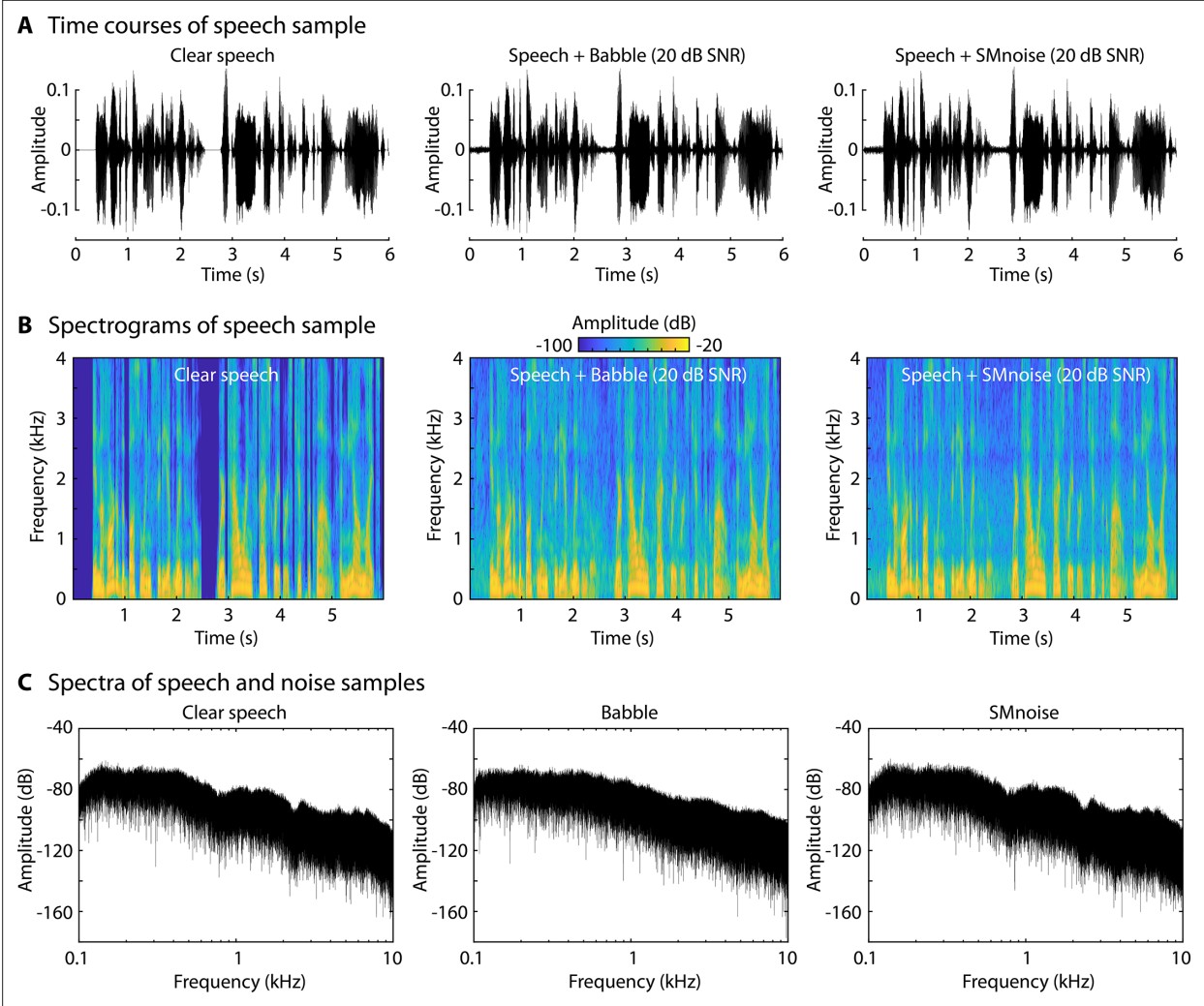

**Figure 3.** Depiction of stimulus samples. (**A**) Time courses for clear speech and speech to which background babble or speech-matched noise was added at 20 dB signal-to-noise ratio (SNR; all sound mixtures were normalized to the same root-mean-square amplitude). The first 6 s of a story are shown. (**B**) Spectrograms of the samples in Panel A. (**C**) Power spectra for clear speech, babble, and speech-matched noise. In **C**, only background babble/noise is displayed, without added speech.

ignoring the stories. EEG was recorded concurrently. Story comprehension and gist were not assessed to avoid that participants feel they should pay attention to the speech materials.

The behavioral results showed no significant differences for hit rate or response time in the visual 1-back task between the clear condition and any of the SNRs ($p_{FDR} > 0.05$; *Figure 2A*).

EEG prediction accuracy did not differ between clear speech and any SNR level ($p_{FDR} > 0.05$; *Figure 3B*). In contrast, and similar to Experiment 1, the P1-N1 amplitude of the TRF was significantly greater for all SNR levels relative to clear speech ($p_{FDR} \leq 0.05$; *Figure 3C, D*). An explorative piece-wise regression revealed a breaking point at +4.4 dB SNR, showing a significant linear increase in P1-N1 amplitude from +28.4 to +4.4 dB SNR ($t_{21} = -3.506$, $p = 0.002$, $d = 0.747$; right to left in *Figure 3D*), whereas the P1-N1 amplitude decreased from +4.4 to −0.4 dB SNR ($t_{21} = 2.416$, $p = 0.025$, $d = 0.515$). No differences were found for the P2-N1 amplitude ($p_{FDR} > 0.05$; *Figure 3C, D*). There was no correlation between visual task performance and the noise-related enhancement in P1-N1 amplitude (*Figure 2—figure supplement 1*), speaking against the possibility that high performers in the visual task expanded effort/attention to speech.

The results of Experiment 2 show that, under diverted attention, neural speech tracking is enhanced for speech presented in background babble at very high SNRs (~30 dB SNR) relative to clear speech. Because participants did not pay attention to the speech in Experiment 2, the results indicate that

attention is unlikely to drive the masker-related enhancement in neural tracking observed here and previously (*Yasmin et al., 2023*; *Panela et al., 2024*). The enhancement may thus be exogenously rather than endogenously driven. Previous work suggests the type of masker may play an important role in whether speech tracking is enhanced by background sound. A masker-related enhancement was reported for a 12-talker babble masker (*Yasmin et al., 2023*; *Panela et al., 2024*), whereas the effect was absent, or relatively small, for a stationary noise that spectrally matched the speech signal (*Ding and Simon, 2013*; *Zou et al., 2019*; *Synigal et al., 2023*). Sound normalization also differed. The work observing the enhancement normalized all SNR conditions to the same overall amplitude (*Yasmin et al., 2023*; *Panela et al., 2024*). This leads to a reduction in the speech-signal amplitude as SNR decreases, which, in fact, works against the observation that neural speech tracking is enhanced for speech in babble. In the other studies, the level of the speech signal was kept the same for all materials and background noise was added at specific SNRs (*Ding and Simon, 2013*; *Synigal et al., 2023*). Using this normalization approach, the overall amplitude of the sound mixture increases with decreasing SNR. Experiment 3 was conducted to disentangle these different potential contributions to the masker-related enhancement in neural speech tracking.

## Experiment 3: Masker-related enhancement of neural tracking is greater for babble than speech-matched noise

In Experiment 3, participants listened to stories in quiet (clear), in background babble, and in noise that spectrally matched the speech signal, both at 10, 15, and 20 dB SNR. Auditory materials were either normalized to the same root-mean-square (RMS) amplitude (i.e., the intensity of the speech signal decreased with increasing SNR; referred to as 'lower intensity' of speech) or the speech signal

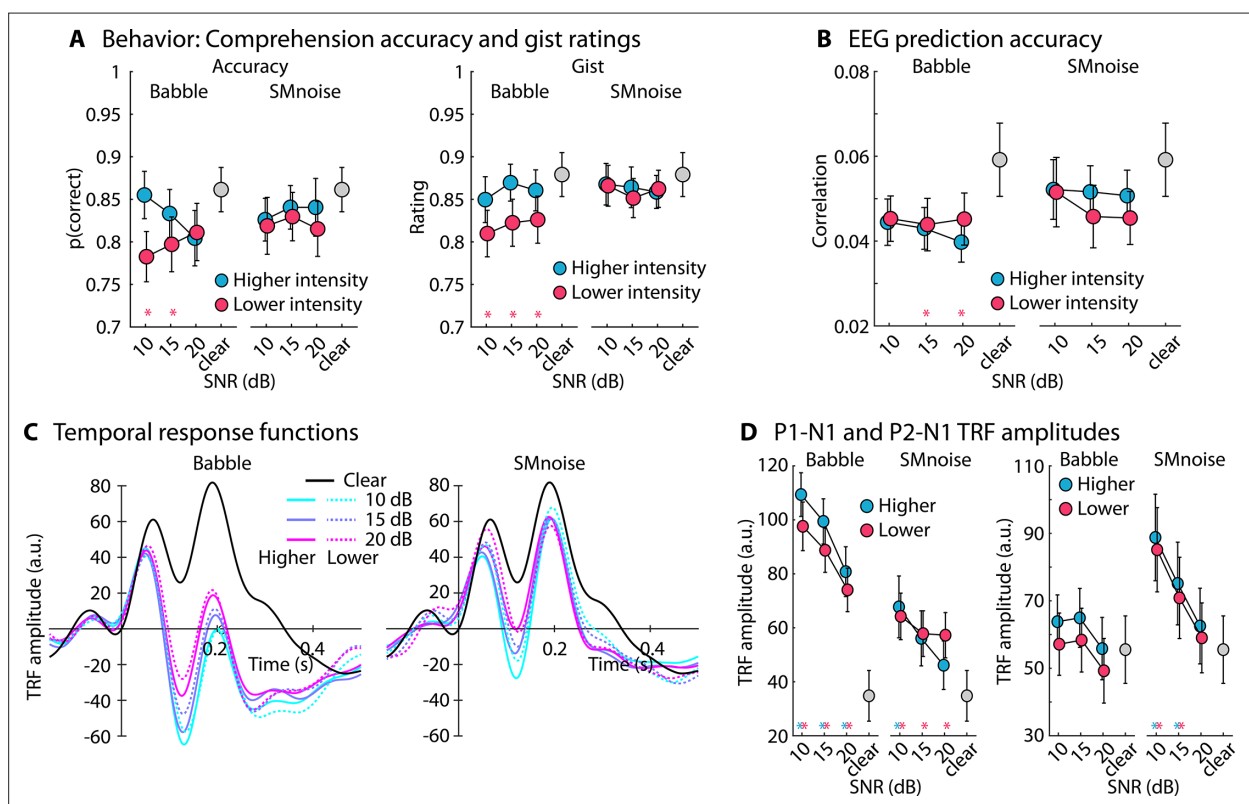

**Figure 4.** Results for Experiment 3 (N=23). (**A**) Accuracy of story comprehension (left) and gist ratings (right). Higher versus lower intensity refers to the two sound-level normalization types, one resulting in a slightly lower intensity of the speech signal in the sound mixture than the other. (**B**) Electroencephalography (EEG) prediction accuracy. (**C**) Temporal response functions (TRFs). (**D**) P1-N1 (left) and P2-N1 (right) amplitude difference for clear speech and different speech-masking and sound normalization conditions. In panels **A, B, and D**, a colored asterisk close to the x-axis indicates a significant difference from a paired t-test relative to the clear condition ($p_{FDR}$ < 0.05; false discovery rate [FDR]-thresholded). The specific color of the asterisk – blue versus red – indicates the normalization type (higher vs. lower speech level, respectively). The absence of an asterisk indicates that there was no significant difference relative to clear speech. Error bars reflect the standard error of the mean.

was kept at the same level across stories and the background sound was added to it at different SNRs (referred to as 'higher intensity' of the speech). Sample time courses, spectrograms, and spectra are shown in *Figure 3*. Participants answered story comprehension questions and rated gist understanding.

The analysis of behavioral performance revealed significantly lower comprehension performance and gist ratings, compared to clear speech, for babble-masked speech that was normalized such that the speech level was slightly lower than for clear speech (*Figure 4A*). The repeated measures analysis of variance (rmANOVA; Masker Type, Normalization Type; excluding clear speech) for the proportion of correctly answered comprehension questions did not reveal any effects or interactions (for all p > 0.1). The rmANOVA for gist ratings revealed an effect of Normalization Type ($F_{1,22}$ = 4.300, p = 0.05, $\omega^2$ = 0.008), showing higher gist ratings when the speech signal had a 'higher' compared to a 'lower' intensity, whereas the other effects and interactions were not significant (for all p > 0.05).

The analysis of EEG prediction accuracy revealed lower accuracies for 15 and 20 dB SNR for the normalization resulting in lower speech intensity relative to clear speech, but otherwise no differences between clear speech and the masked speech (for all $p_{FDR}$ > 0.05; *Figure 4B*). The rmANOVA, focusing on the two factors (Normalization Type, Masker Type), did not reveal any significant effects nor interactions (for all p > 0.25).

The analysis of the TRFs revealed the following results. For the babble masker, the P1-N1 amplitudes were larger for all SNRs compared to clear speech, for both sound-level normalization types (for all $p_{FDR}$ ≤ 0.05). For the speech-matched noise, P1-N1 amplitudes were larger for all SNRs compared to clear speech for the normalization resulting in lower speech intensity (for all $p_{FDR}$ ≤ 0.05), but only for 10 dB SNR for the normalization resulting in higher speech intensity (15 dB SNR was significant for an uncorrected *t*-tests). The rmANOVA revealed larger P1-N1 amplitudes for the 12-talker babble compared to the speech-matched noise masker (effect of Masker Type: $F_{1,22}$ = 32.849, p = 9.1 · $10^{-6}$, $\omega^2$ = 0.162) and larger amplitudes for lower SNRs (effect of SNR: $F_{2,44}$ = 22.608, p = 1.8 · $10^{-7}$, $\omega^2$ = 0.049). None of the interactions nor the effect of Normalization Type were significant (for all p > 0.05).

Analysis of the P2-N1 revealed larger amplitudes for speech masked by speech-matched noise at 10 and 15 dB SNR, for both normalization types (for all $p_{FDR}$ ≤ 0.05). None of the other masked speech conditions differed from clear speech. The rmANOVA revealed an effect of SNR ($F_{2,44}$ = 26.851, p = 2.7 · $10^{-8}$, $\omega^2$ = 0.024), Masker Type ($F_{1,22}$ = 5.859, p = 0.024, $\omega^2$ = 0.023), and an SNR × Masker Type interaction ($F_{2,44}$ = 7.684, p = 0.001, $\omega^2$ = 0.007). The interaction was due to an increase in P2-N1 amplitude with decreasing SNR for the speech-matched noise (all $p_{Holm}$ ≤ 0.05), whereas P2-N1 amplitudes for babble did not differ significantly between SNR conditions (all $p_{Holm}$ > 0.05).

The results of Experiment 3 show that neural speech tracking increases for babble and speech-matched noise maskers compared to clear speech, but that the 12-talker babble masker leads to a greater enhancement compared to the speech-matched noise. Slight variations in the level of the

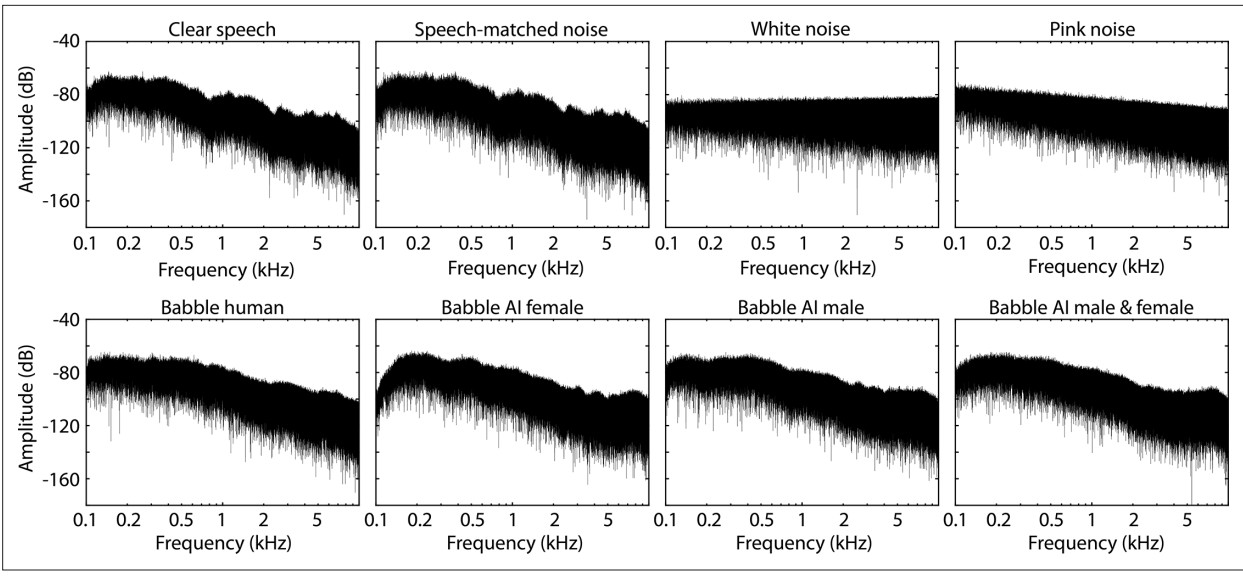

**Figure 5.** Spectra for clear speech and different background noises.

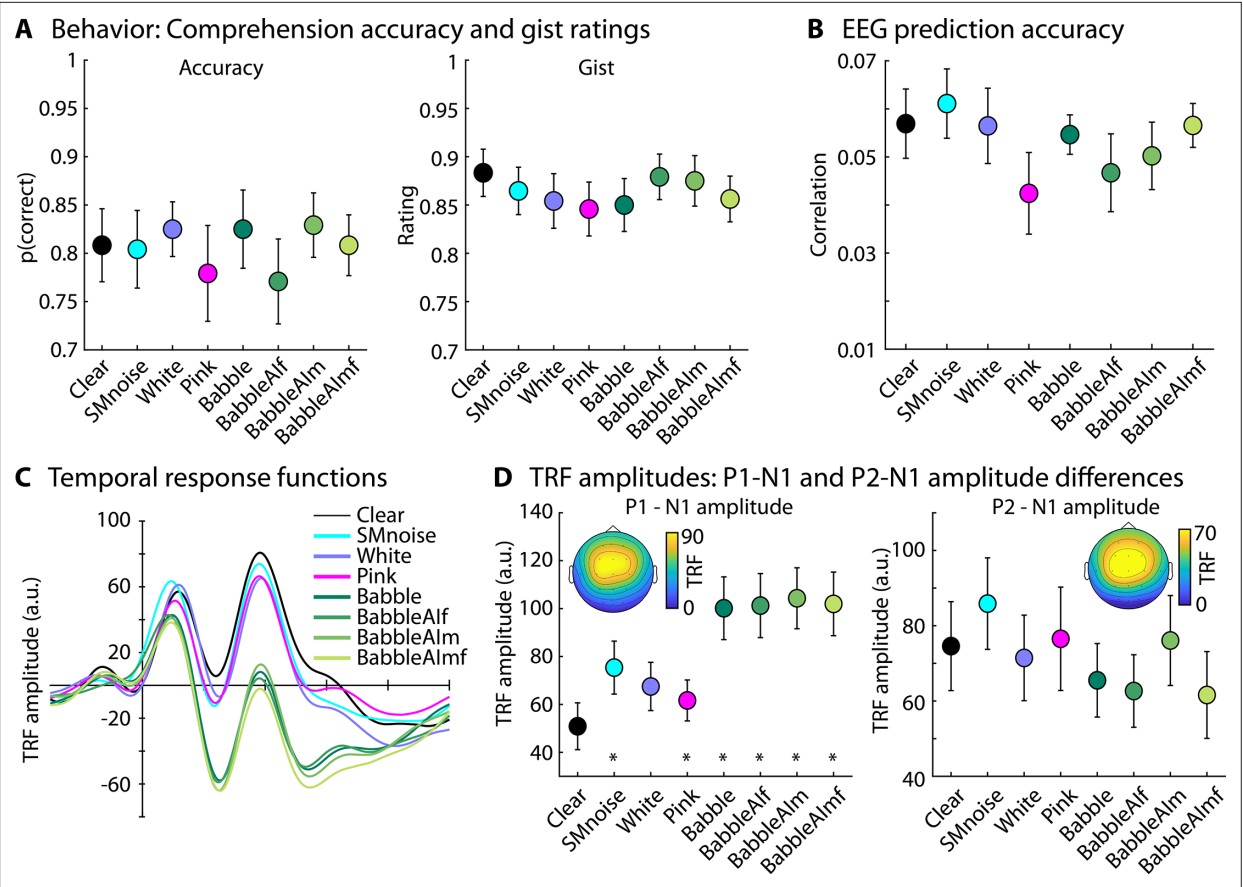

**Figure 6.** Results for Experiment 4 (N=20). (**A**) Accuracy of story comprehension (left) and gist ratings (right). (**B**) Electroencephalography (EEG) prediction accuracy. (**C**) Temporal response functions (TRFs). (**D**) P1-N1 and P2-N1 amplitude difference for clear speech and different speech-masking conditions. Topographical distributions reflect the average across all conditions. In panels A, B, and D, the black asterisk close to the *x*-axis indicates a significant difference from a paired t-test relative to the clear condition ($p_{FDR}$ < 0.05; false discovery rate [FDR]-thresholded). The absence of an asterisk indicates that there was no significant difference relative to clear speech. Error bars reflect the standard error of the mean.

speech signal in the sound mixture (resulting from different sound-level normalization procedures) do not seem to overly impact the results. Because Experiment 3 indicates that the type of background noise may affect the degree of masker-related enhancement, we conducted Experiment 4 to investigate whether different types of commonly used noises lead to similar enhancements in neural speech tracking.

## Experiment 4: Neural-tracking enhancements generalize across different masker types

In Experiment 4, participants listened to stories in quiet (clear) or in the presence of different types of background maskers at +20 dB SNR while EEG was recorded. Masker types included stationary noise that spectrally matched the speech signal (Experiment 3), white noise, pink noise, 12-talker babble (Experiments 1–3), and three newly created 12-talker babbles. The three new babble maskers were used to ensure there is nothing specific about the babble masker used in Experiments 1–3 and our previous work (*Yasmin et al., 2023*; *Panela et al., 2024*) that could lead to an enhanced tracking response. The three new babble maskers varied in spectral properties associated with different voice genders (male, female, male + female) made of 12 speech streams generated using an artificial intelligence (AI)-based speech synthesizer. The amplitude spectra for clear speech and each masker type are shown in *Figure 5*. Participants answered comprehension questions and rated gist understanding.

Comprehension accuracy and gist ratings for the clear story did not significantly differ from the data for masked speech (for all $p_{FDR}$ > 0.05; *Figure 6A*).

EEG prediction accuracy did not significantly differ between clear speech and any of the masker types (for all $p_{FDR} > 0.05$; *Figure 6B*). In contrast, the TRF P1-N1 amplitude was larger for all masker types, expect for white noise, compared to clear speech (for all $p_{FDR} \leq 0.05$; *Figure 6D*; the difference between clear speech and speech masked by white noise was significant when uncorrected; $p = 0.05$). There were no differences among the four different babble maskers (for all $p > 0.6$), indicating that different voice genders of the 12-talker babble do not differentially affect the masker-related enhancement in neural speech tracking. However, the P1-N1 amplitude was larger for speech masked by the babble maskers (collapsed across the four babble maskers) compared to speech masked by white noise ($t_{19} = 4.133$, $p = 5.7 \cdot 10^{-4}$, $d = 0.68$), pink noise ($t_{19} = 5.355$, $p = 3.6 \cdot 10^{-5}$, $d = 1.197$) and the noise that spectrally matched to speech ($t_{19} = 3.039$, $p = 0.007$, $d = 0.68$). There were no differences among the three noise maskers (for all $p > 0.05$). The P2-N1 amplitude for clear speech did not differ from the P2-N1 amplitude for masked speech (for all $p_{FDR} > 0.05$; *Figure 6D*, right).

The results of Experiment 4 replicate the results from Experiments 1–3 by showing that babble noise at a high SNR (20 dB) increases neural speech tracking. Experiment 4 further shows that the neural-tracking enhancement generalizes across different noises, albeit a bit less for white noise (significant only when uncorrected for multiple comparisons). Results from Experiment 4 also replicate the larger tracking enhancement for speech in babble noise compared to speech in speech-matched noise observed in Experiment 3.

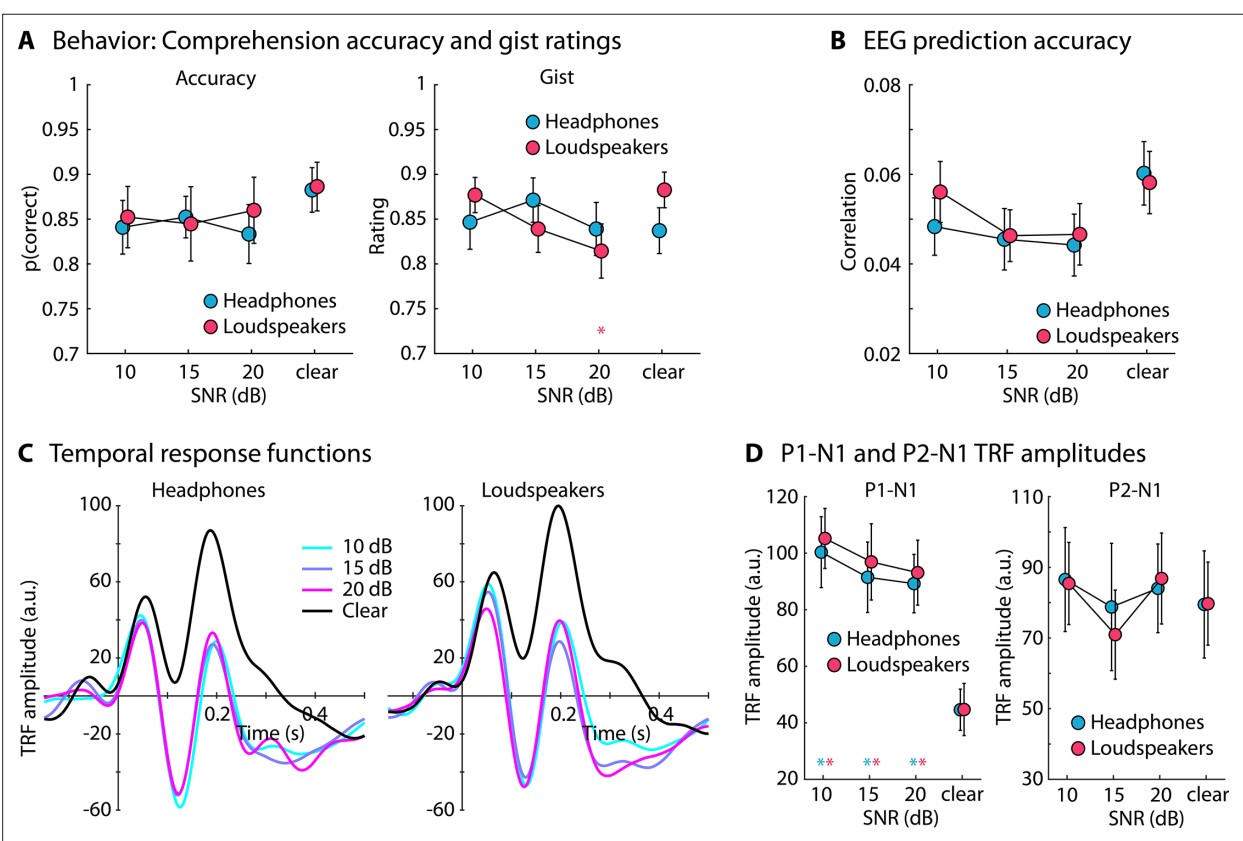

**Figure 7.** Results for Experiment 5 (N=22). (**A**) Accuracy of story comprehension (left) and gist ratings (right). (**B**) Electroencephalography (EEG) prediction accuracy. (**C**) Temporal response functions (TRFs). (**D**) P1-N1 and P2-N1 amplitude difference for clear speech and different speech-masking and sound-delivery conditions. In panels **A, B, and D**, a colored asterisk close to the *x*-axis indicates a significant difference from a paired t-test relative to the clear condition ($p_{FDR} < 0.05$; false discovery rate [FDR]-thresholded). The specific color of the asterisk – blue versus red – indicates the sound-delivery type. The absence of an asterisk indicates that there was no significant difference relative to clear speech. Error bars reflect the standard error of the mean.

## Experiment 5: Neural-tracking enhancement generalizes to free-field listening

Sounds in Experiments 1–4 were presented via headphones, which is comparable to previous work using headphones or in-ear phones (*Alain et al., 2009*; *Alain et al., 2012*; *Ding and Simon, 2013*; *Alain et al., 2014*; *Broderick et al., 2018*; *Decruy et al., 2019*; *Tune et al., 2021*; *Synigal et al., 2023*; *Yasmin et al., 2023*; *Panela et al., 2024*). However, headphones or in-ear phones attenuate external sound sources such that clear speech is arguably presented in 'unnatural' quiet. In everyday life, speech typically reaches our ears in free-field space. Experiment 5 examines whether the noise-related enhancement in neural speech tracking also generalizes to free-field listening. Participants listened to stories in quiet and in 12-talker babble (+10, +15, and +20 dB SNR) either via headphones or loudspeakers.

Comprehension accuracy and gist ratings are shown in *Figure 7A*. There were no differences between clear speech and speech masked by background babble for any of the conditions, with the exception of a lower gist rating for the 20 dB SNR loudspeaker condition (*Figure 7A*, right).

For the EEG prediction accuracy, there were no differences between the clear conditions and any of the masked speech conditions (*Figure 7B*). The rmANOVA (3 SNRs [+10, +15, +20 dB] × 2 Sound Delivery methods) did not reveal effects or interactions involving SNR or Sound Delivery (ps >0.15).

The analysis of TRF amplitudes revealed significantly larger P1-N1 amplitudes for all masked speech conditions compared to clear speech (for all $p_{FDR} \leq 0.05$; *Figure 7D*, left). A rmANOVA (3 SNRs × 2 Sound Delivery methods) did not reveal any effects or interaction involving SNR or Sound Delivery (ps > 0.1). For the P2-N1 amplitude, there were no differences between clear speech and any of the masked speech conditions (for all $p_{FDR} > 0.05$; *Figure 7D*, right). The rmANOVA revealed an effect of SNR ($F_{2,42}$ = 3.953, p = 0.027, $\omega^2$ = 0.005), caused by the lower P2-N1 amplitude for the +15 dB SNR conditions compared to the +10 dB SNR ($t_{42}$ = 2.493, $p_{Holm}$ = 0.05, d = 0.171) and the +20 dB SNR condition ($t_{42}$ = 2.372, $p_{Holm}$ = 0.05, d = 0.162). There was no effect of Sound Delivery nor an SNR × Sound Delivery interaction (ps >0.5).

The results of Experiment 5 show that the enhanced neural tracking of speech associated with minor background babble is unrelated to delivering sounds via headphones (which typically attenuate sounds in the environment). Instead, minor background babble at +20 dB SNR also increased the neural tracking of speech under free-field (loudspeaker) conditions, thus pointing to the generalizability of the phenomenon to conditions more akin to naturalistic listening scenarios.

## Discussion

In five EEG experiments, the current study investigated the degree to which background masking sounds at high SNRs, for which speech is highly intelligible, affect neural speech tracking. The results show that 12-talker babble enhances neural tracking at very high SNRs (~30 dB) relative to clear speech (Experiment 1) and that this enhancement is present even when participants carry out an unrelated visual task (Experiment 2), suggesting that attention or effort do not cause the noise-related neural-tracking enhancement. The results further show that the enhancement of neural speech tracking is greater for speech in the presence of 12-talker babble compared to a stationary noise that spectrally matches the speech (Experiments 3 and 4), although both masker types spectrally overlap with the speech. The enhancement was also greater for 12-talker babble compared to pink noise and white noise (Experiment 4). Finally, the enhanced neural speech tracking generalizes from headphone to free-field listening (Experiment 5), pointing to the real-world nature of the tracking enhancement. Overall, the current study paints a clear picture of a highly generalized enhancement of neural speech tracking in the presence of minimal background noise, making links to speech intelligibility and listening difficulties in noise challenging.

## Enhanced neural tracking of speech under minimal background noise

Across all five experiments of the current study, speech masked by background noise at high SNRs (up to 30 dB SNR) led to enhanced neural tracking of the amplitude-onset envelope of speech (this was also present for the amplitude envelope, see *Figure 1—figure supplement 1*). The enhancement was present for different background maskers, but most prominently for 12-talker babble. Previous work on neural speech tracking also observed enhanced neural tracking for speech masked by 12-talker

babble at moderate SNRs (~12 dB; *Yasmin et al., 2023*; *Panela et al., 2024*), consistent with the current study. The current results are also consistent with studies showing a noise-related enhancement to tone bursts (*Alain et al., 2009*; *Alain et al., 2012*; *Alain et al., 2014*), syllable onsets (*Parbery-Clark et al., 2011*), and high-frequency temporal modulations in sounds (*Ward et al., 2010*; *Shukla and Bidelman, 2021*). Other work, using a noise masker that spectrally matches the target speech, has not reported tracking enhancements (*Ding and Simon, 2013*; *Zou et al., 2019*; *Synigal et al., 2023*). However, in these works, SNRs have been lower (<10 dB) to investigate neural tracking under challenging listening conditions. At low SNRs, neural speech tracking decreases (*Ding and Simon, 2013*; *Zou et al., 2019*; *Yasmin et al., 2023*; *Figures 1 and 2*), thus resulting in an inverted u-shape in relation to SNR for attentive and passive listening (Experiments 1 and 2). Moreover, the speech-tracking enhancement was smaller for speech-matched noise compared to babble noise (*Figures 4 and 6*), potentially explaining the absence of the enhancement for speech-matched noise at low SNRs in previous work (*Ding and Simon, 2013*; *Zou et al., 2019*; *Synigal et al., 2023*).

The noise-related enhancement in the neural tracking of the speech envelope was greatest for 12-talker babble, but it was also present for speech-matched noise, pink noise, and, to some extent, white noise. The latter three noises bare no perceptual relation to speech, but resemble stationary, background buzzing from industrial noise, heavy rain, waterfalls, wind, or ventilation. Twelve-talker babble – which is also a stationary masker – is clearly recognizable as overlapping speech, but words or phonemes cannot be identified (*Bilger, 1984*; *Bilger et al., 1984*; *Wilson, 2003*; *Wilson et al., 2012a*). There may thus be something about the naturalistic, speech nature of the background babble that facilitates neural speech tracking.

The spectral power for both the 12-talker babble and the speech-matched noise overlaps strongly with the spectral properties of the speech signal, although the speech-matched noise most closely resembles the spectrum of speech. Spectral overlap of the background sound and the speech signal could cause energetic masking in the auditory periphery and degrade accurate neural speech representations (*Brungart et al., 2006*; *Mattys et al., 2012*; *Wilson et al., 2012b*; *Kidd et al., 2019*). Although peripheral masking would not explain why neural speech tracking is enhanced in the first place, more peripheral masking for the speech-matched noise compared to the 12-talker babble would be consistent with a reduced enhancement for the former compared to the latter masker. However, pink noise and white noise spectrally overlap much less with speech than the other two background maskers, but the neural-tracking enhancement did not differ between the speech-matched noise, pink noise, and white noise maskers. This again suggests that there may be something about the speech nature of the babble masker that drives the larger neural-tracking enhancement.

Critically, the current results have implications for research and clinical applications. The neural tracking of the speech envelope has been linked to speech intelligibility (*Ding et al., 2014*; *Vanthornhout et al., 2018*; *Lesenfants et al., 2019*) and has been proposed to be a useful clinical biomarker for speech encoding in the brain (*Dial et al., 2020*; *Gillis et al., 2022*; *Schmitt et al., 2022*; *Kries et al., 2024*; *Panela et al., 2024*). However, speech intelligibility, assessed here via gist ratings (*Davis and Johnsrude, 2003*; *Ritz et al., 2022*), only declines for SNRs below 15 dB SNR (consistent with intelligibility scores; *Irsik et al., 2022*), whereas the neural-tracking enhancement is already present for ~30 dB SNR. This result questions the link between neural envelope tracking and speech intelligibility, or at least makes the relationship non-linear (cf. *Yasmin et al., 2023*). Research on the neural tracking of speech using background noise must thus consider that the noise itself may enhance the tracking.

## Potential mechanisms associated with enhanced neural speech tracking

Enhanced neural tracking associated with a stationary background masker or noise-vocoded speech has been interpreted to reflect an attentional gain when listeners must invest cognitively to understand speech (*Hauswald et al., 2022*; *Yasmin et al., 2023*; *Panela et al., 2024*). However, these works used moderate to low SNRs or moderate speech degradations, making it challenging to distinguish between attentional mechanisms and mechanisms driven by background noise per se. The current study demonstrates that attention unlikely causes the enhanced neural tracking of the speech onset-envelope. First, the tracking enhancement was observed for very high SNRs (~30 dB) at which speech is highly intelligible (*Holder et al., 2018*; *Rowland et al., 2018*; *Spyridakou et al., 2020*; *Irsik et al., 2022*). Arguably, little or no effort is needed to understand speech at ~30 dB SNR

(*Rowland et al., 2018*), making attentional gain an unlikely explanation. Importantly, neural speech tracking was enhanced even when participants performed a visual task and were passively presented with the speech materials (*Figure 2*). Participants are unlikely to invest effort to understand speech when performing an attention-demanding visual task. Taken together, the current study provides little evidence that noise-related enhancements of neural speech tracking are due to attention or effort investment.

Another possibility put forward in the context of enhanced neural responses to tone bursts in noise (*Alain et al., 2009*; *Alain et al., 2012*; *Alain et al., 2014*) is that background noise increases arousal, which, in turn, amplifies the neural response to sound. However, a few pieces of evidence are inconsistent with this hypothesis. Arousal to minimal background noise habituates quickly (*Alvar and Francis, 2024*) and arousal does not appear to affect early sensory responses but rather later, non-sensory responses in EEG (>150 ms; *Han et al., 2013*). Moreover, pupil dilation – a measure of arousal (*Mathôt, 2018*; *Joshi and Gold, 2020*; *Burlingham et al., 2022*) – is similar for speech in noise at SNRs ranging from +16 to +4 dB SNR (*Ohlenforst et al., 2017*; *Ohlenforst et al., 2018*), for which neural tracking increases (*Figure 1*). Hence, arousal is unlikely to explain the noise-related enhancement in neural speech tracking, but more direct research is needed to clarify this further.

A third potential explanation of enhanced neural tracking is stochastic resonance, reflecting an automatic mechanism in neural circuits (*Stein et al., 2005*; *McDonnell and Abbott, 2009*; *McDonnell and Ward, 2011*). Stochastic resonance has been described as the facilitation of a near-threshold input through noise. That is, a near-threshold stimulus or neuronal input, that alone may be insufficient to drive a neuron beyond its firing threshold, can lead to neuronal firing if noise is added, because the noise increases the stimulus or neuronal input for brief periods, causing a response in downstream neurons (*Ward et al., 2002*; *Moss et al., 2004*). However, the term is now used more broadly to describe any phenomenon where the presence of noise in a non-linear system improves the quality of the output signal than when noise is absent (*McDonnell and Abbott, 2009*). Stochastic resonance has been observed in humans in several domains, such as in tactile, visual, and auditory perception (*Kitajo et al., 2003*; *Wells et al., 2005*; *Tabarelli et al., 2009*, but see *Rufener et al., 2020*). In the current study, speech was presented at suprathreshold levels, but stochastic resonance may still play a role at the neuronal level (*Stocks, 2000*; *McDonnell and Abbott, 2009*). EEG signals reflect the synchronized activity of more than 10,000 neurons (*Niedermeyer and Silva, 2005*). Some neurons may not receive sufficiently strong input to elicit a response when a person listens to clear speech but may be pushed beyond their firing threshold by the additional, acoustically elicited noise in the neural system. Twelve-talker babble was associated with the greatest noise-related enhancement in neural tracking, possibly because the 12-talker babble facilitated neuronal activity in speech-relevant auditory regions, where the other, non-speech noises were less effective.

## Conclusions

The current study provides a comprehensive account of a generalized increase in the neural tracking of the amplitude-onset envelope of speech due to minimal background noise. The results show that (1) neural speech tracking is enhanced for speech masked by background noise at very high SNRs (~30 dB), (2) this enhancement is independent of attention, (3) it generalizes across different stationary background maskers, although being strongest for 12-talker babble, and (4) it is present for headphone and free-field listening, suggesting the neural-tracking enhancement generalizes to real-life situations. The work paints a clear picture that minimal background noise enhances the neural representation of the speech envelope. The work further highlights the non-linearities of neural speech tracking as a function of background noise, challenging the feasibility of neural speech tracking as a biological marker for speech processing.

## Methods

**Key resources table**

| Reagent type (species) or resource | Designation | Source or reference | Identifiers | Additional information |
|---|---|---|---|---|
| Software, algorithm | MATLAB | MATLAB | RRID:SCR_001622 | |
| Software, algorithm | JASP | JASP | RRID:SCR_015823 | |

*Continued on next page*

*Continued*

| Reagent type (species) or resource | Designation | Source or reference | Identifiers | Additional information |
|---|---|---|---|---|
| Software, algorithm | PsychToolbox | PsychToolbox | RRID:SCR_002881 | |
| Software, algorithm | OpenAI | ChatGPT | RRID:SCR_023775 | |
| Software, algorithm | FieldTrip | FieldTrip | RRID:SCR_004849 | |

## Participants

The current study comprised five experiments. Participants were native English speakers or grew up in English-speaking countries (mostly Canada) and have been speaking English since early childhood (<5 years of age). Participants reported having normal hearing abilities and no neurological disease (one person reported having attention-deficit/hyperactivity disorder, but this did not affect their participation). Participants gave written informed participation consent, consent to publish, and data-sharing consent prior to the experiment and were compensated for their participation. The study was conducted in accordance with the Declaration of Helsinki, the Canadian Tri-Council Policy Statement on Ethical Conduct for Research Involving Humans (TCPS2-2014), and was approved by the Research Ethics Board of the Rotman Research Institute at Baycrest Academy for Research and Education (REB # 21-27).

Twenty-two adults participated in Experiment 1 (median: 23.5 years; range: 18–35 years; 12 males, 9 females, 1 transgender). Twenty-two adults participated in Experiment 2 (median: 23 years; range: 18–31 years; 11 males, 10 females, 1 transgender; five additional datasets were excluded due to low performance in the visual task; see details below). Twenty-three people participated in Experiment 3 (median: 25 years; range: 19–33 years; 7 males, 15 females, 1 transgender; data from one additional person were excluded due to technical issues resulting in missing triggers). Twenty individuals participated in Experiment 4 (median: 25.5 years; range: 19–34 years; 4 males, 15 females, 1 transgender; one additional dataset was excluded due to missing triggers). Twenty-two adults participated in Experiment 5 (median: 26 years; range: 19–34 years; 10 males, 11 females, 1 transgender; one additional dataset was excluded due to spurious triggers and artifacts). Several participants took part in more than one of the experiments, in separate sessions on separate days: 7, 7, 9, 9, and 14 (for Experiments 1–5, respectively) participated only in one experiment; 3 individuals participated in all 5 experiments; 68 unique participants took part across the 5 experiments.

## Sound environment and stimulus presentation

Data collection was carried out in a sound-attenuating booth. Sounds were presented via Sennheiser (HD 25-SP II) headphones and computer loudspeakers (Experiment 5) through an RME Fireface 400 external sound card. Stimulation was run using Psychtoolbox (v3.0.14, RRID:SCR_002881) in MATLAB (MathWorks Inc, RRID:SCR_001622) on a Lenovo T480 laptop with Microsoft Windows 7. Visual stimulation was projected into the sound booth via screen mirroring. All sounds were presented at about 65 dB SPL.

## Story materials

In each of the 5 experiments, participants listened to 24 stories of about 1:30 to 2:30 min duration each. OpenAI's GPT chat (*OpenAI, 2023*; RRID:SCR_023775) was used to generate each story, five corresponding comprehension questions, and four associated multiple-choice answer options (1 correct, 3 incorrect). Each story was on a different topic (e.g., a long-lost friend, a struggling artist). GPT stories, questions, and answer choices were manually edited wherever needed to ensure accuracy. Auditory story files were generated using Google's modern AI-based speech synthesizer using the male 'en-US-Neural2-J' voice with default pitch, speed, and volume parameters (i.e., 0, 1, and 0, respectively; https://cloud.google.com/text-to-speech/docs/voices). Modern AI speech is experienced as very naturalistic and speech perception is highly similar for AI and human speech (*Herrmann, 2023*). A new set of 24 stories and corresponding comprehension questions and multiple-choice options were generated for each of the 5 experiments (due to a mistake 8 stories from Experiments 1–4 were also used in Experiment 5). The duration of silences between sentences was about 1 s in

the stories generated through Google's synthesizer, which sounded artificially long. They were thus shortened to 0.6 s to sound more naturalistic.

After each story in Experiments 1, 3, 4, and 5, participants answered the five comprehension questions about the story. Each comprehension question comprised four response options (chance level = 25%). Participants further rated the degree to which they understood the gist of what was said in the story, using a 9-point scale that ranged from 1 (strongly disagree) to 9 (strongly agree; the precise wording was: 'I understood the gist of the story' and 'Please rate this statement independently of how you felt about the other stories)'. Gist ratings were linearly scaled to range between 0 and 1 to facilitate interpretability similar to the proportion of correct responses (*Mathiesen et al., 2024*; *Panela et al., 2024*). For short sentences, gist ratings have been shown to highly correlated with speech intelligibility scores (*Davis and Johnsrude, 2003*; *Ritz et al., 2022*). In Experiment 2, participants performed a visual *n*-back task (detailed below) while stories were presented and no comprehension questions nor the gist rating were administered.

## EEG recordings and preprocessing

Electroencephalographical signals were recorded from 16 scalp electrodes (Ag/Ag–Cl-electrodes; 10–20 placement) and the left and right mastoids using a BioSemi system (Amsterdam, The Netherlands). The sampling frequency was 1024 Hz with an online low-pass filter of 208 Hz. Electrodes were referenced online to a monopolar reference feedback loop connecting a driven passive sensor and a common-mode-sense active sensor, both located posteriorly on the scalp.

Offline analysis was conducted using MATLAB software (MathWorks Inc, RRID:SCR_001622). An elliptic filter was used to suppress power at the 60 Hz line frequency (stopband 59.5–60.5 Hz, 80 dB suppression). Data were re-referenced by averaging the signal from the left and right mastoids and subtracting the average separately from each of the 16 channels. Re-referencing to the averaged mastoids was calculated to gain high SNR for auditory responses at fronto-central-parietal electrodes (*Ruhnau et al., 2012*; *Herrmann et al., 2013*). Data were filtered with a 0.7-Hz high-pass filter (length: 2449 samples, Hann window) and a 22-Hz low-pass filter (length: 211 samples, Kaiser window).

EEG data were segmented into time series time-locked to story onset and down-sampled to 512 Hz. Independent components analysis (FieldTrip toolbox, *Oostenveld et al., 2011*, RRID:SCR_004849) was used to remove signal components reflecting blinks, eye movement, and noise artifacts (*Bell and Sejnowski, 1995*; *Makeig et al., 1995*; *Oostenveld et al., 2011*). After the independent components analysis, remaining artifacts were removed by setting the voltage for segments in which the EEG amplitude varied more than 80 µV within a 0.2-s period in any channel to 0 µV (cf. *Dmochowski et al., 2012*; *Dmochowski et al., 2014*; *Cohen and Parra, 2016*; *Irsik et al., 2022*; *Yasmin et al., 2023*; *Panela et al., 2024*). Data were low-pass filtered at 10 Hz (251 points, Kaiser window) because neural signals in the low-frequency range are most sensitive to acoustic features (*Di Liberto et al., 2015*; *Zuk et al., 2021*; *Yasmin et al., 2023*).

## Calculation of amplitude-onset envelopes

For each clear story (i.e., without background babble or noise), a cochleogram was calculated using a simple auditory-periphery model with 30 auditory filters (*McDermott and Simoncelli, 2011*; cutoffs evenly spaced on the ERB scale; *Glasberg and Moore, 1990*). The resulting amplitude envelope for each auditory filter was compressed by 0.6 to simulate inner ear compression (*McDermott and Simoncelli, 2011*). Such a computationally simple peripheral model has been shown to be sufficient, as compared to complex, more realistic models, for envelope-tracking approaches (*Biesmans et al., 2017*). Amplitude envelopes were averaged across auditory filters and low-pass filtered at 40 Hz filter (Butterworth filter, fourth order). To obtain the amplitude-onset envelope, the first derivative was calculated and all negative values were set to zero (*Hertrich et al., 2012*; *Fiedler et al., 2017*; *Fiedler et al., 2019*; *Yasmin et al., 2023*; *Panela et al., 2024*). The onset-envelope was down-sampled to match the sampling of the EEG data. The amplitude-onset envelope was selected because (1) several previous works have used it (*Hertrich et al., 2012*; *Fiedler et al., 2017*; *Brodbeck et al., 2018a*; *Daube et al., 2019*; *Fiedler et al., 2019*), (2) our previous work first observing the enhancement also used the amplitude-onset envelope (*Yasmin et al., 2023*; *Panela et al., 2024*), and (3) the amplitude-onset envelope has been suggested to elicit a strong speech tracking response (*Hertrich et al., 2012*).

Results for analyses using the amplitude envelope instead of the amplitude-onset envelope show similar effects and are provided in the Supplementary Materials (*Figure 1—figure supplement 1*).

## EEG TRF and prediction accuracy

A forward model based on the linear TRF (*Crosse et al., 2016*; *Crosse et al., 2021*) was used to quantify the relationship between the amplitude-onset envelope of a story and EEG activity (note that cross-correlation led to very similar results, *Figure 1—figure supplement 2*; cf. *Hertrich et al., 2012*). The ridge regularization parameter lambda ($\lambda$), which prevents overfitting, was set to 10 based on previous work (*Fiedler et al., 2017*; *Fiedler et al., 2019*; *Yasmin et al., 2023*; *Panela et al., 2024*). Pre-selection of $\lambda$ based on previous work avoids extremely low and high $\lambda$ on some cross-validation iterations and avoids substantially longer computational time. Pre-selection of $\lambda$ also avoids issues if limited data per condition are available, as in the current study (*Crosse et al., 2021*).

For each story, 50 25 s data snippets (*Crosse et al., 2016*; *Crosse et al., 2021*) were extracted randomly from the EEG data and corresponding onset-envelope (*Panela et al., 2024*). Each of the 50 EEG and onset-envelope snippets were held out once as a test dataset, while the remaining non-overlapping EEG and onset-envelope snippets were used as training datasets. Overlapping snippets in the training data were used to increase the amount of data in the training given the short duration of the stories. Speech-clarity levels were randomized across stories and all analyses were conducted similarly for all conditions. Hence, no impact of overlapping training data on the results is expected (consistent with noise-related enhancements observed previously when longer stories and non-overlapping data were used; *Yasmin et al., 2023*). Analyses using cross-correlation, for which data snippets are treated independently, show similar results compared to those reported here using TRFs (*Figure 1—figure supplement 2*).

For each training dataset, linear regression with ridge regularization was used to map the onset-envelope onto the EEG activity to obtain a TRF model for lags ranging from 0 to 0.4 s (*Hoerl and Kennard, 1970*; *Crosse et al., 2016*; *Crosse et al., 2021*). The TRF model calculated for the training data was used to predict the EEG signal for the held-out test dataset. The Pearson correlation between the predicted and the observed EEG data of the test dataset was used as a measure of EEG prediction accuracy (*Crosse et al., 2016*; *Crosse et al., 2021*). Model estimation and prediction accuracy were calculated separately for each of the 50 data snippets per story, and prediction accuracies were averaged across the 50 snippets.

EEG prediction accuracy was calculated because many previous studies report it (e.g., *Decruy et al., 2019*; *Broderick et al., 2021*; *Gillis et al., 2021*; *Weineck et al., 2022*; *Karunathilake et al., 2023*), but the main focus of the current study is on the TRF weights/amplitude. That is, to investigate the neural-tracking response directly, we calculated TRFs for each training dataset for a broader set of lags, ranging from –0.15 to 0.5 s, to enable similar analyses as for traditional event-related potentials (*Yasmin et al., 2023*; *Panela et al., 2024*). TRFs were averaged across the 50 training datasets and the mean in the time window –0.15 to 0 s was subtracted from the data at each time point (baseline correction).

Data analyses focused on a fronto-central electrode cluster (F3, Fz, F4, C3, Cz, and C4) known to be sensitive to neural activity originating from auditory cortex (*Näätänen and Picton, 1987*; *Picton et al., 2003*; *Herrmann et al., 2018*; *Irsik et al., 2021*). Prediction accuracies and TRFs were averaged across the electrodes of this fronto-central electrode cluster prior to further analysis.

Analyses of the TRF focused on the P1-N1 and the P2-N1 amplitude differences. The amplitude of individual TRF components (P1, N1, and P2) was not analyzed because the TRF time courses for the clear condition had an overall positive shift (see also *Yasmin et al., 2023*; *Panela et al., 2024*) that could bias analyses more favorably toward response differences which may, however, be harder to interpret. The P1, N1, and P2 latencies were estimated from the averaged time courses across participants, separately for each SNR. P1, N1, and P2 amplitudes were calculated for each participant and condition as the mean amplitude in the 0.02-s time window centered on the peak latency. The P1-minus-N1 and P2-minus-N1 amplitude differences were calculated.

## Statistical analyses

All statistical analyses were carried out using MATLAB (MathWorks Inc, RRID:SCR_001622) and JASP software (*JASP, 2023*; version 0.18.3.0, RRID:SCR_015823).

## Experiment 1: Stimuli, procedures, and analyses

Participants listened to 24 stories in 6 blocks (4 stories per block). Three of the 24 stories were played under clear conditions (i.e., without background noise). Twelve-talker babble was added to the other 21 stories (*Bilger, 1984*; *Bilger et al., 1984*; *Wilson et al., 2012a*). Twelve-talker babble is a standardized masker in speech-in-noise tests (*Bilger, 1984*; *Bilger et al., 1984*) that simulates a crowded restaurant, while not permitting the identification of individual words in the masker (*Mattys et al., 2012*). The babble masker was added at SNRs ranging from +30 to –2 dB in 21 steps of 1.6 dB SNR. Speech in background babble at 15–30 dB SNR is highly intelligible (*Holder et al., 2018*; *Rowland et al., 2018*; *Spyridakou et al., 2020*; *Irsik et al., 2022*). No difference in speech intelligibility during story listening has been found between clear speech and speech masked by a 12-talker babble at +12 dB SNR (*Irsik et al., 2022*). Intelligibility typically drops below 90% of correctly reported words for +7 dB and lower SNR levels (*Irsik et al., 2022*). Hence, listeners have no trouble understanding speech at the highest SNRs used in the current study. All speech stimuli were normalized to the same RMS amplitude and presented at about 65 dB SPL. Participants listened to each story, and after each story rated gist understanding and answered comprehension questions. Stories were presented in random order. Assignment of speech-clarity levels (clear speech and SNRs) to specific stories was randomized across participants.

Behavioral data (comprehension accuracy, gist ratings), EEG prediction accuracy, and TRFs for the three clear stories were averaged. For the stories in babble, a sliding average across SNRs was calculated for behavioral data, EEG prediction accuracy, and TRFs, such that data for three neighboring SNRs were averaged. Averaging across three stories was calculated to reduce noise in the data and match the averaging of three stories for the clear condition. For TRFs, analyses focused on the P1-N1 and the P2-N1 amplitude differences. For the statistical analyses, the clear condition was compared to each SNR (resulting from the sliding average) using a paired samples *t*-test. False discovery rate (FDR) was used to account for multiple comparisons (*Benjamini and Hochberg, 1995*; *Genovese et al., 2002*). In cases where the data indicated a breaking point in behavior or brain response as a function of SNR, an explorative piece-wise regression (broken-stick analysis) with two linear pieces was calculated (*McZgee and Carleton, 1970*; *Vieth, 1989*; *Toms and Lesperance, 2003*). Identification of the breaking point and two pieces was calculated on the across-participant average as the minimum root-mean-squared error. A linear function was then fit to each participant's data as a function of SNR and the estimated slope was tested against zero using a one-sample *t*-test, separately for each of the two pieces.

## Experiment 2: Stimuli, procedures, and analyses

A new set of 24 stories was generated and participants were presented with 4 stories in each of 6 blocks. Speech-clarity levels were the same as in Experiment 1 (i.e., clear speech and SNRs ranging from +30 to –2 dB SNR). In Experiment 2, participants were instructed to ignore the stories and instead perform a visual 1-back task. In no part of the experiment were participants instructed to attend to the speech.

For the visual 1-back task, images of white digits (0–9) on black background were taken from the MNIST Handwritten Digit Classification Dataset (*Li Deng, 2012*). The digit images were selected, because different images of the same digit differ visually and thus make it challenging to use a simple feature-matching strategy to solve the 1-back task. A new digit image was presented every 0.5 s throughout the time over which a story was played (1:30 to 2:30 min). A digit image was presented for 0.25 s followed by a 0.25-s black screen before the next image was presented. The continuous stream of digits contained a digit repetition (albeit a different image sample) every 6–12 digits. Participants were tasked with pressing a button on a keyboard as soon as they detected a repetition. We did not include comprehension questions or gist ratings in Experiment 2 to avoid that participants feel they should pay attention to the speech materials. Hit rate and response time were used as behavioral measures.

Analyses examining the effects of SNR in Experiment 2 were similar to the analyses in Experiment 1. Behavioral data (hit rate, response time in 1-back task) and EEG data (prediction accuracy, TRFs) for the three clear stories were averaged and a sliding average procedure across SNRs (three neighbors) was used for stories in babble. Statistical tests compared SNR conditions to the clear condition, including FDR-thresholding. An explorative piece-wise regression with two linear pieces

was calculated in cases where the data indicated a breaking point in behavior or brain response as a function of SNR. Data from five participants were excluded from analysis because they performed below 60% in the visual *n*-back task. A low performance could mean that the participants did not fully attend to the visual task and instead attended to the spoken speech. To avoid this possibility, data from low performers were excluded.

## Experiment 3: Stimuli, procedures, and analyses

A new set of 24 stories, comprehension questions, and multiple-choice options were generated. Participants listened to four stories in each of six blocks. Four of the 24 stories were presented under clear conditions. Ten stories were masked by 12-talker babble at five different SNRs (two stories each: +5, +10, +15, +20, and +25 dB SNR), whereas the other 10 stories were masked by a stationary noise that spectrally matched the speech signal at five different SNRs (two stories each: +5, +10, +15, +20, and +25 dB SNR). To obtain the spectrally matched noise, the long-term spectrum of the clear speech signal was calculated using a fast Fourier transform (FFT). The inverted FFT was calculated using the frequency-specific amplitudes estimated from the speech signal jointly with randomly selected phases for each frequency. *Figure 3* shows a time course snippet, a power spectrum, and a short snippet of the spectrogram for clear speech, speech masked by babble, and speech masked by the spectrally matched noise. Twelve-talker babble and speech-matched noise spectrally overlap extensively (*Figure 3*), but 12-talker babble recognizably contains speech (although individual elements cannot be identified), whereas the speech-matched noise does not contain recognizable speech elements.

For 5 of the 10 stories per masker type (babble, noise), the speech-plus-masker sound signal (mixed at a specific SNR) was normalized to the same RMS amplitude as the clear speech stories. This normalization results in a decreasing level of the speech signal in the speech-plus-masker mixture as SNR decreases. For the other five stories, the RMS amplitude of the speech signal was kept the same for all stories and a masker was added at the specific SNRs. This normalization results in an increasing sound level (RMS) of the sound mixture as SNR decreases. In other words, the speech signal in the sound mixture is played at a slightly lower intensity for the former than for the latter normalization type. We thus refer to this manipulation as speech with 'lower' and 'higher' intensities for the two normalization types, respectively. Note that these differences in speech level are very minor due to the high SNRs used here. Stories were presented in randomized order and the assignment of stories to conditions was randomized across participants.

Behavioral data (comprehension accuracy, gist ratings), EEG prediction accuracy, and TRFs for the four clear stories were averaged. For the stories in babble and speech-matched noise, a sliding average across SNRs was calculated for behavioral data, EEG prediction accuracy, and TRFs, such that data for four neighboring SNR levels were averaged, separately for the two masker types (babble, noise) and normalization types (adjusted speech level, non-adjusted speech level). Averaging across four stories was calculated to reduce noise in the data and match the number of stories included in the average for the clear condition. For TRFs, analyses focused on the P1-N1 and the P2-N1 amplitude differences. For the statistical analyses, the clear condition was compared to each SNR level (resulting from the sliding average) using a paired samples *t*-test. FDR was used to account for multiple comparisons (*Benjamini and Hochberg, 1995*; *Genovese et al., 2002*). Differences between masker types and normalization types were examined using an rmANOVA with the within-participant factors SNR (10, 15, and 20 dB SNR), Masker Type (babble, speech-match noise), and Normalization Type (lower vs. higher speech levels). Post hoc tests were calculated for significant main effects and interactions. Holm's methods was used to correct for multiple comparisons (*Holm, 1979*). Statistical analyses were carried out in JASP software (v0.18.3; *JASP, 2023*, RRID:SCR_015823). Note that JASP uses pooled error terms and degrees of freedom from an rmANOVA for the corresponding post hoc effects. The reported degrees of freedom are thus higher than for direct contrasts had they been calculated independently from the rmANOVA.

## Experiment 4: Stimuli, procedures, and analyses

A new set of 24 stories, corresponding comprehension questions, and multiple-choice options were generated. Participants listened to four stories in each of six blocks. After each story, they answered five comprehension questions and rated gist understanding. Three stories were presented in each of eight conditions: clear speech (no masker), speech with added white noise, pink noise, stationary

noise that spectrally matched the speech signal (Experiment 3), 12-talker babble (Experiments 1–3), and three additional 12-talker babbles. The additional babble maskers were created to ensure there is nothing specific about the babble masker used in our previous work (*Yasmin et al., 2023*; *Panela et al., 2024*) and in Experiments 1–3 that could lead to an enhanced tracking response and to vary spectral properties of the babble associated with different voice genders (male, female). For the three additional babble maskers, 24 text excerpts of about 800 words each were taken from Wikipedia (e.g., about flowers, forests, insurance, etc.). The 24 text excerpts were fed into Google's AI speech synthesizer to generate 24 continuous speech materials (~5 min) of which 12 were from male voices and 12 from female voices. The three 12-talker babble maskers were created by adding speech from 6 male and 6 female voices (mixed gender 12-talker babble), speech from the 12 male voices (male gender 12-talker babble), and speech from the 12 female voices (female gender 12-talker babble). Maskers were added to the speech signal at 20 dB SNR and all acoustic stimuli were normalized to the same RMS amplitude. A power spectrum for each of the seven masker types is displayed in *Figure 5*. Stories were presented in randomized order and the assignment of stories to the eight different conditions was randomized across participants.

Behavioral data (comprehension accuracy, gist ratings), EEG prediction accuracy, and TRFs were averaged across the three stories for each condition. For TRFs, analyses focused on the P1-N1 and the P2-N1 amplitude differences. For the statistical analyses, the clear condition was compared to the masker conditions using a paired samples *t*-test. FDR was used to account for multiple comparisons (*Benjamini and Hochberg, 1995*; *Genovese et al., 2002*). Differences between the four different babble maskers, between babble and noise maskers, and between the three noise maskers were also investigated using paired samples *t*-tests.

## Experiment 5: Stimuli, procedures, and analyses

A new set of 24 stories, comprehension questions, and multiple-choice options were generated (due to a mistake 8 stories from Experiments 1–4 were eventually used in Experiment 5; and 16 new stories). Participants listened to four stories in each of six blocks. After each story, they answered five comprehension questions and rated gist understanding. For three of the six blocks, participants listened to the stories through headphones, as for Experiments 1–4, whereas for the other three blocks, participants listened to the stories via computer loudspeakers placed in front of them. Blocks for different sound-delivery conditions (headphones, loudspeakers) alternated within each participant's session, and the starting condition was counter-balanced across participants. For each sound-delivery condition, participants listened to three stories each under clear conditions, +10, +15, and +20 dB SNR (12-talker babble, generated using Google's AI voices as described for Experiment 4). Stories were distributed such that the four speech-clarity conditions were presented in each block in randomized order. All acoustic stimuli were normalized to the same RMS amplitude. The sound level of the headphones and the sound level of the loudspeakers (at the location of a participant's head) were matched.

Behavioral data (comprehension accuracy, gist ratings), EEG prediction accuracy, and TRFs were averaged across the three stories for each speech-clarity and sound-delivery condition. For TRFs, analyses focused on the P1-N1 and the P2-N1 amplitude differences. For the statistical analyses, the clear condition was compared to the masker conditions using a paired samples *t*-test. FDR was used to account for multiple comparisons (*Benjamini and Hochberg, 1995*; *Genovese et al., 2002*). To test for differences between sound-delivery types, an rmANOVA was calculated, using the within-participant factors SNR (10, 15, and 20 dB SNR; clear speech was not included, because a difference to clear speech was tested directly as just described) and Sound Delivery (headphones, loudspeakers). Post hoc tests were calculated using dependent samples *t*-tests, and Holm's methods were used to correct for multiple comparisons (*Holm, 1979*).

## Acknowledgements

We thank Priya Pandey, Tiffany Lao, and Saba Junaid for their help with data collection. The research was supported by the Canada Research Chair program (CRC-2019-00156) and the Natural Sciences and Engineering Research Council of Canada (Discovery Grant: RGPIN-2021-02602).

# Additional information

## Funding

| Funder | Grant reference number | Author |
|---|---|---|
| Canada Research Chairs | CRC-2019-00156 | Björn Herrmann |
| Natural Sciences and Engineering Research Council of Canada | RGPIN-2021-02602 | Björn Herrmann |

The funders had no role in study design, data collection, and interpretation, or the decision to submit the work for publication.

## Author contributions

Björn Herrmann, Conceptualization, Resources, Data curation, Software, Formal analysis, Supervision, Funding acquisition, Validation, Investigation, Visualization, Methodology, Writing – original draft, Project administration, Writing – review and editing

## Author ORCIDs

Björn Herrmann  https://orcid.org/0000-0001-6362-3043

## Ethics

Participants gave written informed participation consent, consent to publish, and data-sharing consent prior to the experiment and were compensated for their participation. The study was conducted in accordance with the Declaration of Helsinki, the Canadian Tri-Council Policy Statement on Ethical Conduct for Research Involving Humans (TCPS2-2014), and was approved by the Research Ethics Board of the Rotman Research Institute at Baycrest Academy for Research and Education (REB # 21-27).

Reviewer #1 (Public review): https://doi.org/10.7554/eLife.100830.3.sa1
Reviewer #2 (Public review): https://doi.org/10.7554/eLife.100830.3.sa2
Author response https://doi.org/10.7554/eLife.100830.3.sa3

# Additional files

## Supplementary files

MDAR checklist

## Data availability

Data and analysis code are available at https://osf.io/zs9u5/.

The following dataset was generated:

| Author(s) | Year | Dataset title | Dataset URL | Database and Identifier |
|---|---|---|---|---|
| Herrmann B | 2024 | Enhanced neural speech tracking through noise indicates stochastic resonance in humans | https://osf.io/zs9u5/ | Open Science Framework, zs9u5 |

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
