## [Editor Report · eLife Assessment]

This study presents an **important** contribution to the understanding of neural speech tracking, demonstrating how minimal background noise can enhance the neural tracking of the amplitude-onset envelope. The evidence, through a well-designed series of EEG experiments, is **convincing**. This work will be of interest to auditory scientists, particularly those investigating biological markers of speech processing.

---

## [Referee Report · Reviewer #1 (Public review)]

This paper presents a comprehensive study of how neural tracking of speech is affected by background noise. Using five EEG experiments and Temporal response function (TRF), it investigates how minimal background noise can enhance speech tracking even when speech intelligibility remains very high. The results suggest that this enhancement is not attention-driven but could be explained by stochastic resonance. These findings generalize across different background noise types, listening conditions, and speech features (envelope onset and envelope), offering insights into speech processing in real-world environments.

I find this paper well-written, the experiments and results are clearly described.

Comments on revisions:

I thank the author for thoughtful revisions and for adequately addressing my comments. The new version is much clearer and improved. I have no further questions.

---

## [Referee Report · Reviewer #2 (Public review)]

The author investigates the role of background noise on EEG-assessed speech tracking in a series of five experiments. In the first experiment the influence of different degrees of background noise is investigated and enhanced speech tracking for minimal noise levels is found. The following four experiments explore different potential influences on this effect, such as attentional allocation, different noise types and presentation mode.

The step-wise exploration of potential contributors to the effect of enhanced speech tracking for minimal background noise is compelling. The motivation and reasoning for the different studies is clear and logical and therefore easy to follow. The results are discussed in a concise and clear way. While I specifically like the conciseness, one inevitable consequence is that not all results are equally discussed in depth.

Based on the results of the five experiments, the authors conclude that the enhancement of speech tracking for minimal background noise is likely due to stochastic resonance. Given broad conceptualizations of stochasitc resonance as noise benefit this is a reasonable conclusion.

This study will likely impact the field as it provides compelling support questioning the relationship between speech tracking and speech processing.

Comments on revisions:

All my previous comments were addressed nicely. Some of the comments were mere curiosity questions that were nicely entertained, even though they were not of direct relevance to the manuscript. I like the addition of the amplitude envelope analysis to the supplementary material as it offers direct comparison of those different methods. My only tiny tiny critic is (which bears no significance), that due to the many rearrangement changes in the marked changes document, the changes of content get buried and hard to see.

---

## [Author Response]

The following is the authors’ response to the original reviews.

**Reviewer #1 (Public review):**
This paper presents a comprehensive study of how neural tracking of speech is a ected by background noise. Using five EEG experiments and Temporal response function (TRF), it investigates how minimal background noise can enhance speech tracking even when speech intelligibility remains very high. The results suggest that this enhancement is not attention-driven but could be explained by stochastic resonance. These findings generalize across di erent background noise types and listening conditions, o ering insights into speech processing in real-world environments. I find this paper well-written, the experiments and results are clearly described. However, I have a few comments that may be useful to address.

I thank the reviewer for their positive feedback.

(1) The behavioral accuracy and EEG results for clear speech in Experiment 4 di er from those of Experiments 1-3. Could the author provide insights into the potential reasons for this discrepancy? Might it be due to linguistic/ acoustic di erences between the passages used in experiments? If so, what was the rationale behind using di erent passages across di erent experiments?

The slight di erences in behavior and EEG magnitudes may be due to several factors. Di erent participants took part in the di erent experiments (with some overlap). Stories and questions were generated using ChatGPT using the same approach, but di erent research assistants have supported story and question generation, and ChatGPT advanced throughout the course of the study, such that di erent versions were used over time (better version control was only recently introduced by OpenAI). The same Google voice was used for all experiments, so this cannot be a factor. Most critically, within each experiment, assignment of speech-clarity conditions to di erent stories was randomized, such that statistical comparisons are una ected by these minor di erences between experiments. The noise-related enhancement generalizes across all experiments, showing that minor di erences in experimental materials do not impact it.

(2) Regarding peak amplitude extraction, why were the exact peak amplitudes and latencies of the TRFs for each subject not extracted, and instead, an amplitude average within a 20 ms time window based on the group-averaged TRFs used? Did the latencies significantly di er across di erent SNR conditions?

Estimation of peak latency can be challenging if a deflection is not very pronounced in a participant. Especially the N1 was small for some conditions. Using the mean amplitude in a specific time window is very common practice in EEG research that mitigates this issue. Another, albeit less common, approach is to use a Jackknifing procedure to estimate each participant’s latencies (Smulders 2010 Psychophysiology; although this may sometimes not work well). For the revision, I used the Jackknifing approach to estimate peak latencies for each participant and condition, and extracted the mean amplitude around the peak latency. As expected, this approach provides very similar e ects as reported in the main article, here exemplified for Experiments 1 and 2. The results are thus not a ected by this data analysis choice. The estimated latencies di ered across SNRs, e.g., the N1 increased with decreasing SNR (this is less surprising/novel and was thus not added to the manuscript to avoid increasing the amount of information).

**Author response image 1. sa3fig1:** P1-minus-N1 amplitude for Experiment 1 and 2, using amplitudes centered on individually estimated peak latencies. The asterisk indicates a significant di erence from the clear speech condition (FDR-thresholded).

(3) How is neural tracking quantified in the current study? Does improved neural tracking correlate with EEG prediction accuracy or individual peak amplitudes? Given the di ering trends between N1 and P2 peaks in babble and speech-matched noise in experiment 3, how is it that babble results in greater envelope tracking compared to speech-matched noise?

Neural tracking is generally used for responses resulting from TRF analyses, crosscorrelations, or coherence, where the speech envelope is regressed against the brain signals (see review of Brodbeck & Simon 2020 Current Opinion in Physiology). Correlations between EEG prediction accuracy and individual peak amplitudes was not calculated because the data used for the analyses are not independent. The EEG prediction accuracy essentially integrates information over a longer time interval (here 0–0.4 s), whereas TRF amplitudes are more temporally resolved. If one were to shorten the time interval (e.g., 0.08–0.12 s), then EEG prediction accuracy would look more similar to the TRF results (because the TRF is convolved with the amplitude-onset envelope of the speech [predicted EEG] before calculating the EEG prediction accuracy). Regarding the enhancement di erence between speech-matched noise and babble, I have discussed a possible interpretation in the discussion section. The result is indeed surprising, but it replicates across two experiments (Experiments 3 and 4), and is consistent with previous work using speech-matched noise that did not find the enhancement. I reproduce the part of the discussion here.

“Other work, using a noise masker that spectrally matches the target speech, have not reported tracking enhancements (Ding and Simon, 2013; Zou et al., 2019; Synigal et al., 2023). However, in these works, SNRs have been lower (<10 dB) to investigate neural tracking under challenging listening conditions. At low SNRs, neural speech tracking decreases (Ding and Simon, 2013; Zou et al., 2019; Yasmin et al., 2023; Figures 1 and 2), thus resulting in an inverted u-shape in relation to SNR for attentive and passive listening (Experiments 1 and 2).”

“The noise-related enhancement in the neural tracking of the speech envelope was greatest for 12talker babble, but it was also present for speech-matched noise, pink noise, and, to some extent, white noise. The latter three noises bare no perceptional relation to speech, but resemble stationary, background buzzing from industrial noise, heavy rain, waterfalls, wind, or ventilation. Twelve-talker babble – which is also a stationary masker – is clearly recognizable as overlapping speech, but words or phonemes cannot be identified (Bilger, 1984; Bilger et al., 1984; Wilson, 2003; Wilson et al., 2012b). There may thus be something about the naturalistic, speech nature of the background babble that facilitates neural speech tracking.”

“Twelve-talker babble was associated with the greatest noise-related enhancement in neural tracking, possibly because the 12-talker babble facilitated neuronal activity in speech-relevant auditory regions, where the other, non-speech noises were less e ective.”

(4) The paper discusses how speech envelope-onset tracking varies with di erent background noises. Does the author expect similar trends for speech envelope tracking as well? Additionally, could you explain why envelope onsets were prioritized over envelope tracking in this analysis?

The amplitude-onset envelope was selected because several previous works have used the amplitude-onset envelope, our previous work that first observed the enhancement also used the amplitude-onset envelope, and the amplitude-onset envelope has been suggested to work better for speech tracking. This was added to the manuscript. For the manuscript revision, analyses were calculated for the amplitude envelope, largely replicating the results for the amplitude-onset envelope. The results for the amplitude envelope are now presented in the Supplementary Materials and referred to in the main text.

“The amplitude-onset envelope was selected because (a) several previous works have used it (Hertrich et al., 2012; Fiedler et al., 2017; Brodbeck et al., 2018a; Daube et al., 2019; Fiedler et al., 2019), (b) our previous work first observing the enhancement also used the amplitude-onset envelope (Yasmin et al., 2023; Panela et al., 2024), and (c) the amplitude-onset envelope has been suggested to elicit a strong speech tracking response (Hertrich et al., 2012). Results for analyses using the amplitude envelope instead of the amplitude-onset envelope show similar e ects and are provided in the Supplementary Materials (Figure 1-figure supplement 1).”

**Recommendations for the authors:**
(1) Include all relevant parameters related to data analysis where applicable. For example, provide the filter parameters (Line 154, Line 177, Line 172), and the default parameters of the speech synthesizer (Line 131).

Additional filter information and parameter values are provided in the revised manuscript.

(2) Please share the data and codes or include a justification as to why the data cannot be shared.

Data and code are provided on OSF (https://osf.io/zs9u5/). A materials availability statement has been added to the manuscript.

**Reviewer #2 (Public review):**
The author investigates the role of background noise on EEG-assessed speech tracking in a series of five experiments. In the first experiment, the influence of di erent degrees of background noise is investigated and enhanced speech tracking for minimal noise levels is found. The following four experiments explore di erent potential influences on this e ect, such as attentional allocation, di erent noise types, and presentation mode. The step-wise exploration of potential contributors to the e ect of enhanced speech tracking for minimal background noise is compelling. The motivation and reasoning for the di erent studies are clear and logical and therefore easy to follow. The results are discussed in a concise and clear way. While I specifically like the conciseness, one inevitable consequence is that not all results are equally discussed in depth. Based on the results of the five experiments, the author concludes that the enhancement of speech tracking for minimal background noise is likely due to stochastic resonance. Given broad conceptualizations of stochastic resonance as a noise benefit this is a reasonable conclusion. This study will likely impact the field as it provides compelling support questioning the relationship between speech tracking and speech processing.

I thank the reviewer for the positive review and thoughtful feedback.

**Recommendations for the authors:**
As mentioned in the public review, I like the conciseness. However, some points might benefit from addressing them.(1) The absence of comprehension e ects is on the one hand surprising, as the decreased intelligibility should (theoretically) be visible in this data. On the other hand, from my own experience, the generation of "good" comprehension questions is quite di icult. While it is mentioned in the methods section, that comprehension accuracy and gist rating go hand in hand, this is not the case here. I am wondering if the data here should be rather understood as "there is no di erence in intelligibility" or that comprehension assessment via comprehension questions is potentially not a valid measure.

I assume that the reviewer refers to Experiment 1, where SNRs approximately below 15 dB led to reduced gist ratings (used as a proxy for speech intelligibility; Davis and Johnsrude, 2003, J Neurosci; Ritz et al., 2022, J Neurosci). That story comprehension accuracy does not decrease could be due to the comprehension questions themselves (as indicated by the reviewer, “good” questions can be hard to generate, potentially having low sensitivity). On the other hand, speech for the most di icult SNR was still ‘reasonably’ intelligible (gist ratings suggest ~85% of words could be understood), and participants may still have been able to follow the thread of the story. I do not further discuss this point in the manuscript, since it is not directly related to the noise-related enhancement in the neural tracking response, because the enhancement was present for high SNRs for which gist ratings did not show a di erence relative to clear speech (i.e., 20 dB and above).

(2) However, if I understood correctly, the "lower" manipulation (same RMS for the whole sound stimulus) of experiment 3 was, what was also used in experiment 1. In experiment 3, unlike 1, there are comprehension e ects. I wondered if there are ideas about why that is.

Yes indeed, the ‘lower’ manipulation in Experiment 3 was also used in Experiments 1, 2, 4, and 5. The generation of the stimulus materials was similar across experiments. However, a new set of stories and comprehension questions was used for each experiment and the participants di ered as well (with some overlap). These aspects may have contributed to the di erence.

(3) Concerning the prediction accuracy, for a naive reader, some surrounding information would be helpful: What is the purpose/expectation of this measure? Is it to show that all models are above chance?

EEG prediction accuracy was included here, mainly because it is commonly used in studies using TRFs. A reader may wonder about EEG prediction accuracy if it were not reported. The hypotheses of the current study are related to the TRF weights/amplitude. This was added to the manuscript.

“EEG prediction accuracy was calculated because many previous studies report it (e.g., Decruy et al., 2019; Broderick et al., 2021; Gillis et al., 2021; Weineck et al., 2022; Karunathilake et al., 2023), but the main focus of the current study is on the TRF weights/amplitude.”

(4) Regarding the length of training and test data I got confused: It says per story 50 25-s snippets. As the maximum length of a story was 2:30 min, those snippets were mostly overlapping, right? It seems that depending on the length of the story and the "location within the time series" of the snippets, the number of remaining non-over-lapping snippets is variable. Also, within training, the snippets were overlapping, correct? Otherwise, the data for training would be too short. Again, as a naive reader, is this common, or can overlapping training data lead to overestimations?

The short stories made non-overlapping windows not feasible, but the overlap unlikely a ects the current results. Using cross-correlation (Hertrich et al 2012 Psychophysiology; which is completely independent for di erent snippets) instead of TRFs shows the same results (now provided in the supplementary materials). In one of our previous studies where the enhancement was first observed (Yasmin et al. 2023 Neuropsychologia), non-overlapping data were used because the stories were longer. This makes any meaningful impact of the overlap very unlikely. Critically, speech-clarity levels were randomized and all analyses were conducted in the same way for all conditions, thus not confounding any of the results/conclusions. The methods section was extended to further explain the choice of overlapping data snippets.

“Speech-clarity levels were randomized across stories and all analyses were conducted similarly for all conditions. Hence, no impact of overlapping training data on the results is expected (consistent with noise-related enhancements observed previously when longer stories and non-overlapping data were used; Yasmin et al., 2023). Analyses using cross-correlation, for which data snippets are treated independently, show similar results compared to those reported here using TRFs (Figure 1figure supplement 2).”

(5) For experiment 1, three stories were clear, while the other 21 conditions were represented by one story each. Presumably, the ratio of 3:1 can a ect TRFs?

TRFs were calculated for each story individually and then averaged across three stories: either three clear stories, or three stories in babble for neighboring SNRs. Hence, the same number of TRFs were averaged for clear and noise conditions, avoiding exactly this issue. This was described in the methods section and is reproduced here:

“Behavioral data (comprehension accuracy, gist ratings), EEG prediction accuracy, and TRFs for the three clear stories were averaged. For the stories in babble, a sliding average across SNR levels was calculated for behavioral data, EEG prediction accuracy, and TRFs, such that data for three neighboring SNR levels were averaged. Averaging across three stories was calculated to reduce noise in the data and match the averaging of three stories for the clear condition.”

(6) Was there an overlap in participants?

Some participants took part in several of the experiments in separate sessions on separate days. This was added to the manuscript.

“Several participants took part in more than one of the experiments, in separate sessions on separate days: 7, 7, 9, 9, and 14 (for Experiments 1-5, respectively) participated only in one experiment; 3 individuals participated in all 5 experiments; 68 unique participants took part across the 5 experiments.”

(7) Can stochastic resonance also explain inverted U-shape results with vocoded speech?

This is an interesting question. Distortions to the neural responses to noise-vocoding may reflect internal noise, but this would require additional research. For example, the Hauswald study (2022 EJN), showing enhancements due to noise-vocoding, used vocoding channels that also reduced speech intelligibility. The study would ideally be repeated with a greater number of vocoding channels to make sure the e ects are not driven by increased attention due to reduced speech intelligibility. I did not further discuss this in detail in the manuscript as it would go too far away from the experiments of the current study.

(8) Typo in the abstract: box sexes is probably meant to say both sexes?

This text was removed, because more detailed gender identification is reported in the methods, and the abstract needed shortening to meet the eLife guidelines.

**Reviewing Editor Comments:**
Interesting series of experiments to assess the influence of noise on cortical tracking in di erent conditions, interpreting the results with the mechanism of stochastic resonance.

I thank the editor for their encouraging feedback.

For experiment 2, the author wishes to exclude the role of attention, by making participants perform a visual task. Data from low performers on the visual task was excluded, to avoid that participants attended the spoken speech. However, from the high performers on the visual task, how can you be sure that they did not pay attention to the auditory stimuli as well (as auditory attention is quite automatic, and these participants might be good at dividing their attention)? I understand that you can not ask participants about the auditory task during the experiment, but did you ask AFTER the experiment whether they were able to understand the stimuli? I think this is crucial for your interpretation.

Participants were not asked whether they were able to understand the stimuli. Participants would unlikely invest e ort/attention in understanding the stories in babble without a speech-related task. Nevertheless, for follow-up analyses, I removed participants who performed above 0.9 in the visual task (i.e., the high performers), and the di erence between clear speech and speech in babble replicates. In the plots, data from all babble conditions above 15 dB SNR (highly intelligible) were averaged, but the results look almost identical if all SNRs are averaged. Moreover, the correlation between visual task performance and the babble-related enhancement was not-significant. These analyses were added to the Supplementary Materials (Figure 2-figure supplement 1).

Statistics: inconsistencies across experiments with a lot of simple tests (FDR corrected) and in addition sometimes rmANOVA added - if interactions in rmANOVA are not significant then all the simple tests might not be warranted. So a bit of double dipping and over-testing here, but on the whole the conclusions do not seem to be overstated.

The designs of the di erent experiments di ered, thus requiring di erent statistical approaches. Moreover, the di erent tests assess di erent comparisons. For all experiments, contrasting the clear condition to all noise conditions was the main purpose of the experiments. To correct for multiple comparison, the False Discovery Rate correction was used. Repeated-measures ANOVAs were conducted in addition to this – excluding the clear condition because it would not fit into a factorial structure (e.g., Experiment 3) or to avoid analyzing it twice (e.g., Experiment 5) – to investigate di erences between di erent noise conditions. There was thus no over-testing in the presented study.

Small points:Question on methods: For each story, 50 25-s data snippets were extracted (Page 7, line 190). As you have stories with a duration of 1.5 to 2 minutes, does that mean there is a lot of overlap across data snippets? How does that influence the TRF/prediction accuracy?

The short stories made non-overlapping windows not feasible, but the overlap unlikely a ects the current results. Using cross-correlation (Hertrich et al 2012 Psychophysiology; which is completely independent for di erent snippets) instead of TRFs shows the same results (newly added Figure 1-figure supplement 2). In one of our previous studies where the enhancement was first observed (Yasmin et al. 2023 Neuropsychologia), non-overlapping data were used because the stories were longer. This makes any meaningful impact of the overlap very unlikely. Critically, speechclarity levels were randomized and all analyses were conducted in the same way for all conditions, thus not confounding any of the results/conclusions. The methods section was extended to further explain the choice of overlapping data snippets.

“Overlapping snippets in the training data were used to increase the amount of data in the training given the short duration of the stories. Speech-clarity levels were randomized across stories and all analyses were conducted similarly for all conditions. Hence, no impact of overlapping training data on the results is expected (consistent with noise-related enhancements observed previously when longer stories and non-overlapping data were used; Yasmin et al., 2023). Analyses using crosscorrelation, for which data snippets are treated independently, show similar results compared to those reported here using TRFs (Figure 1-figure supplement 2).”

Results Experiment 3: page 17, line 417: no di erences were found between clear speech and masked speech - is this a power issue (as it does look di erent in the figure, Figure 4b)?

I thank the editor for pointing this out. Indeed, I made a minor mistake. Two comparisons were significant after FDR-thresholding. This is now included in the revised Figure 4. I also made sure the mistake was not present for other analyses; which it was not.